# Learning Universal Sample Difficulty with Pathology Foundation Models in Histopathology Image Analysis

## Abstract

The fast scaling speed of histopathology datasets allows researchers to train various foundation models for disease-centered research with applications in classifying disease-state information and predicting gene expression levels. However, it has been shown that current models tend to be overconfident and make classification at a low-calibration level. This case is underexplored for regression-type tasks such as gene expression prediction as well, which could seriously affect the diagnosis and treatment based on the developed models. To resolve this critical issue, we propose a universal framework[1] to estimate the sample difficulty (USD) in both regression and classification tasks. In particular, we fit the data in the embedding space with Gaussian distribution and then utilize prior-informed relative Mahalanobis distance to estimate sample difficulty. Moreover, we incorporate such difficulty as a weight to regularize the model prediction, which can improve model performance by emphasizing challenging samples. Our method can be seamlessly extended to regression tasks by the incorporation of discrete targets. Extensive experiments demonstrate that our proposed USD can improve the disease-state classification accuracy by up to 3.8% and gene-level correlation by up to 62.2% compared with the most frequently used approaches. Finally, we provide comprehensive ablation tests to demonstrate the importance of including sample difficulty in the training stage and case studies for the reasonability of assigning samples with different difficulty levels.

## 1 Introduction

The analysis of gigapixel-level whole-slide images (WSIs) is an important topic in computational pathology Song et al. (2023a); Bera et al. (2019); Niazi et al. (2019); Al-Janabi et al. (2012). Due to the complexity and scarcity of pathology data, it is difficult for a pathologist to make accurate diagnoses. While machine-learning-based methods have been applied for pathology analysis Neto et al. (2024); Shaban et al. (2024), these models are usually trained with limited data and knowledge, which might not be useful for general purposes Zhang and Metaxas (2024). To solve this issue, extensive efforts have been made to collect large-scale pathology data, bringing in several pathology foundation models (PFM) pre-trained with pathology image or multimodal data Chen et al. (2024b); Lu et al. (2023); Xu et al. (2024a); Ma et al. (2024). Those PFMs generate robust representations for WSIs in either patch level or slide level, which demonstrate state-of-the-art (SOTA) performances for a wide range of tasks including disease-state classification, disease sub-type identification, medical text-image retrieval, etc. Recent research has also explored cases of using features from PFMs to predict gene expressions from hematoxylin and eosin (H&E)–stained images Jia et al. (2024); Xie et al. (2024); Anonymous (2024); Lee et al. (2024b), revealing the potentials of PFMs in handling regression-oriented problems.

Despite this great progress, we often detect misclassified samples in both training and testing sets when using PFMs for classification-oriented problems. The potential reasons could be multifaceted such as assigning wrong labels, changing brightness, adding medical annotation, etc. Given the importance

---

[1]Full codes can be found here: https://anonymous.4open.science/r/USD-13EB/ (also in supplementary files).

of diagnostic accuracy for patients Niazi et al. (2019), handling extensive noise in pathology data is highly essential. Although some researchers have investigated the difficulty of training samples in general image datasets (e.g., ImageNet Deng et al. (2009)) with technique development Cui et al. (2023) on relative Mahalanobis distance Mahalanobis (2018) and data distillation Wang et al. (2024), we have not yet found any research that systematically investigates how to process these difficult samples in pathology images. Moreover, most of the current research on sample difficulty focuses on classification-oriented problems and attempts to improve models with enhanced generalization ability Cui et al. (2023), but how to extend the learning of sample difficulty in regression-oriented problems remains unsolved. For spatial transcriptomic data analysis, predicting gene expression information based on the H&E image is also an emerging field, as the measurement of spatial transcriptomics data is expensive Anonymous (2025); Zeng et al. (2022) for large-scale analysis. In addition, multi-modal information can provide more insights for pathology analysis Qiao et al. (2022), and thus predicting transcriptomics as a new modality allows us to perform additional analyses such as survival prediction Jaume et al. (2024b) and cell-cell communication inference Armingol et al. (2021). Since we find that these expression predictors might fail for certain genes or spots, we plan to dive deeper for an interpretable solution. Therefore, a general framework for understanding and interpreting sample difficulty for pathology image analysis will be extremely helpful for domain experts in the medical field.

In this paper, we propose a **U**niversal Learning Framework for Estimating **S**ample **D**ifficulty (USD) and improving the capacity of PFMs in histopathology image analysis. Different from previous research Cui et al. (2023); Agarwal et al. (2022); Zhu et al. (2024), our method first transfers the concept of sample difficulty into an outlier detection problem, and then models the training difficulty of samples by integrating the prior information jointly with modified relative Mahalanobis distance (*MRMD*). Furthermore, we leverage discrete targets to extend our sample difficulty to the gene expression prediction task, resulting in a universal model for both regression and classification problems. With these novel designs, USD demonstrates a SOTA performance in both disease-state classification and disease sub-type identification across three datasets of different scales. In addition, USD improves the prediction of gene expression levels from the perspectives of both performance and interpretability across eight datasets from different tissues and diseases.

We further visualize the sample difficulty estimated by USD in Figure 1 and perform clustering analysis in Appendix 8.1. Regarding the disease-state classification task, we can observe an intuitive difference in pathological morphology between the selected samples. We also cannot detect the squamous-like regions enriched with cancer cells in the difficult samples labeled as lung squamous cell carcinoma (LUSC). Regarding the gene expression prediction task, we find that the patches with lower cell enrichment or clear tissue patterns are marked with a high difficulty level, which aligns with their Pearson correlation coefficient (PCC) scores. In contrast, for regions with more useful morphological information, these samples are assigned with lower difficulty, which can be validated by accurate predictions. Overall, our method can help

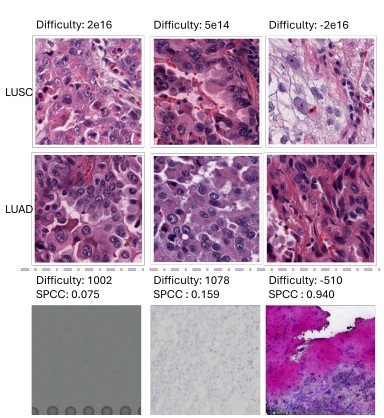

Figure 1: Illustration of sample difficulty (lower means easier).

researchers to better select pathological areas for clinical analysis and filter out useless information.

Our contributions are: (1) we show that PFMs provide superior slide- and patch-level representations, whose features can also estimate sample difficulty; (2) we introduce *MRMD*, a metric for measuring difficulty in classification and regression with fewer false positives; (3) we demonstrate that combining difficulty-aware learning with entropy regularization improves performance; and (4) we design a difficulty-aware loss that boosts results on over 70% of datasets. Beyond these, we establish novelty by conducting the most comprehensive evaluation of PFMs to date across diverse datasets, tasks, and metrics, proving the generalizability of our solutions beyond UNI and surpassing traditional visual models like ResNet50. We further show, for the first time, that PFM features extend naturally to sample difficulty estimation, adding interpretability to pathology workflows. Finally, ablation and robustness analyses clarify when and why PFMs are essential, linking feature representation with difficulty estimation to advance both regression and classification tasks within a unified framework.

## 2 RELATED WORK

**Pathology Foundation Models (PFMs).** Learning robust representations of pathology images is a challenge with extensive applications in computer-aided diagnosis, and PFMs are developed to resolve it. Most of the current PFMs are visual-based or textual-visual-based large-scale neural networks built based on transformer blocks. Moreover, these models diversify in model architectures, pre-training strategy, and training datasets. For example, models such as UNI Chen et al. (2024b) rely on DINOv2 Oquab et al. (2024) as base architecture and Mass-100K dataset in the pre-training stage, while models like GigaPath Xu et al. (2024a) is built based on ViT Dosovitskiy et al. (2021) and utilizes private datasets which are not publicly available. Furthermore, models such as PLIP Huang et al. (2023), CONCH Lu et al. (2024), MUSK Xiang et al. (2025), and TITAN Ding et al. (2024) utilize multi-modal information in the pre-training stage, which enlarges the models' capacity in handling the cross-modality tasks. There also exist models focusing on introducing more modalities in the pre-training stage, such as mSTAR Xu et al. (2024b) with transcriptomic data, as explorations for new pre-training frameworks.

**PFM Applications.** Foundation Models are named after their powerful and wide-ranging downstream capabilities in few-shot and zero-shot learning scenarios, and this is no exception for PFMs. The proposed PFMs have already demonstrated strong abilities in handling disease-related classification tasks, such as disease-state prediction, disease sub-type identification, and image-image retrieval Chen et al. (2024b); Ochi et al. (2024); Xiang and Zhang (2023). These challenges are constrained by data quality and disease heterogeneity and thus they did not have general solutions in the past. Furthermore, PFMs with language capacity can also be applied to addressing multi-modal tasks such as text-image retrieval Huang et al. (2023), visual question answer (VQA) testing Xiang et al. (2025), and medical report generation Shaikovski et al. (2024); Liu et al. (2025b). Recently, researchers also explored the capacity of predicting spot-level gene expression information directly from the paired image information with features obtained from PFMs, which shows potential to help analyze spatial transcriptomics data with lower cost than performing data sequencing directly Anonymous (2024); Lee et al. (2024b). The validation of prediction performances is usually based on databases Jaume et al. (2024a); Chen et al. (2024a) with paired spatial transcriptomics and H&E images.

**Sample Difficulty.** The measurement of sample difficulty can come from either task-specific designs and models Agarwal et al. (2022); Baldock et al. (2021); Zhu et al. (2024), or from pre-trained models Cui et al. (2023). Previously, researchers focused on uncertainty regularization as an effective approach to reducing the overfitting and over-confidence problems in the training stage of the classifier. In the classification problem, most of them are based on the modification of loss functions, for example, Focal loss Liu et al. (2020), $L_p$ norm Joo and Chung (2021), Poly loss (Poly) Leng et al. (2022), Entropy Regularization (ER) Mnih (2016), Weighted Entropy Regularization (WER), and Weighted Poly Loss (WPoly) Cui et al. (2023) are based on adding regularization terms in the loss function to improve the optimization process. The weight could come from the pre-defined distance used to measure the difficulty level of training samples. Other methods such as label smoothing Müller et al. (2019) and correctness ranking loss (CRL) Moon et al. (2020) modify the labels to penalize the samples with the highest prediction confidence, which could be potential solutions. In the regression problem, ordinary entropy (OE) Zhang et al. (2023) is developed to regularize neural networks for handling regression-based tasks inspired by the phenomenon that formatting regression problems as classification problems is helpful. Modified loss functions such as Huber loss Huber (1992) can contribute for reducing the drawbacks caused by underfitting extreme samples.

## 3 METHOD

**Problem Definition.** In this paper, we are given a histopathology dataset $\mathcal{D} = \{\mathbf{x}_i, y_i\}_{i=1}^n$, where $\mathbf{x}_i$ represents an $m$-dimensional feature vector extracted from PFMs for the $i$-th whole-slide image (WSI) or patch (which is an image extracted from WSI based on certain rules) and $y_i$ represents the corresponding targets for prediction, i.e., disease states for the classification task ($y_i$ is a scalar) or gene expression levels ($y_i$ is a vector as we have multiple genes to predict) for regression task. For the classification task, we train a classifier $C_\theta$ based on the training dataset, and may observe sample $\mathbf{x}_d$ whose predicted labels mismatch with the observed label ($C_\theta(\mathbf{x}_d) \neq y_d$). These samples can be treated as difficult samples. Our target is to identify difficult samples and further improve model

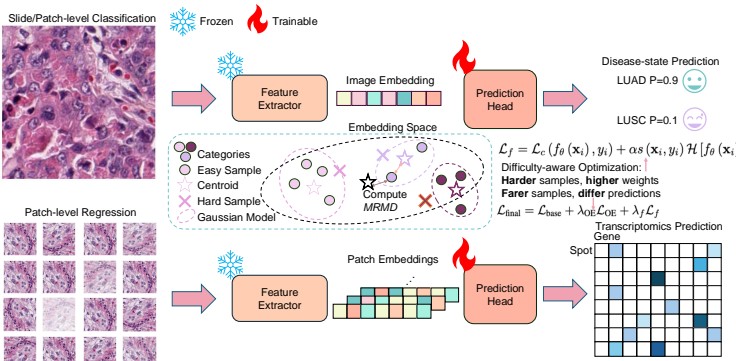

Figure 2: Illustration of USD's pipeline. We accept either slide-level information or patch-level information as input and incorporate the estimated sample difficulty from prior in the training process by reconstructing the target of optimization. By default, PFMs are frozen and only used for extracting image embeddings, while task-specific adapters are trained for different datasets.

performances by correctly predicting these samples in the training stage as many as possible. The formal definition of sample difficulty analysis for the regression problem is similar, and the label of each sample can be computed by discretizing $y$ into different bins, while the mismatched samples are still difficult samples under this context.

**Overview.** USD starts from estimating the sample difficulty levels based on image features extracted from pre-trained base models such as PFMs. We then leverage the sample difficulty to regularize the model outputs in the training stage, as a more difficult sample should be assigned to having a higher weight. To effectively predict gene expression levels based on spatial transcriptomics and paired sets of patches, we consider both sample difficulty and the relationship between expression-level similarity and feature-level similarity. The illustration of USD is shown in Figure 2.

**Foundation Models as Feature Extractor.** We first utilize pre-trained PFMs to embed the images into feature space, which can provide better representations discussed in the previous work Cui et al. (2023). In summary, PFMs are generally trained to ignore low-level information (e.g., class labels) and prioritize whole-image level information rather than low-level image statistics. Moreover, PFMs are trained with more diverse data, which can better learn and extract the intrinsic features of input images and remove noisy information. Therefore, the generated features will be helpful to estimate training difficulty in a robust space and support USD to perform downstream applications.

**Estimating Sample Difficulty with Prior Knowledge.** For training dataset $\mathcal{D}_{train} = \{\mathbf{x}_i, y_i\}_{i=1}^{n_1}$, we first derive the relative Mahalanobis distance (RMD) as the sample difficulty score, which has been shown as a more powerful approach to detect difficult samples Cui et al. (2023). The computation of RMD is introduced later and it can measure the distribution-level difference to define easy samples and difficult samples. For samples with $y_i = k$, we fit a Gaussian model of the set of features $\{\mathbf{x}_i\}$ as $G(\mathbf{x}_i)$. The model can be computed based on:

$$\mathbb{P}(G_k(\mathbf{x}) \mid y = k) = \mathcal{N}\left(G_k(\mathbf{x}) \mid \boldsymbol{\mu}_k, \Sigma\right), \boldsymbol{\mu}_k = \frac{1}{N_k} \sum_{i:y_i=k} G_k\left(\mathbf{x}_i\right),$$

$$\Sigma = \frac{1}{N} \sum_k \sum_{i:y_i=k} \left(G_k\left(\mathbf{x}_i\right) - \boldsymbol{\mu}_k\right)\left(G_k\left(\mathbf{x}_i\right) - \boldsymbol{\mu}_k\right)^\top, \tag{1}$$

where $\boldsymbol{\mu}_k$ represents the mean vector and $\Sigma$ represents the sample covariance matrix, $N_k$ represents the samples belonging to class $k$, and $G_k$ represents the Gaussian model for the class $k$. Similarly, considering all training samples as a background, we can fit a Gaussian model $G_b$:

$$\mathbb{P}(G_b(\mathbf{x})) = \mathcal{N}\left(G_b(\mathbf{x}) \mid \boldsymbol{\mu}_b, \Sigma_b\right), \boldsymbol{\mu}_b = \frac{1}{N} \sum G_b\left(\mathbf{x}_i\right),$$

$$\Sigma_b = \frac{1}{N} \sum \left(G_b\left(\mathbf{x}_i\right) - \boldsymbol{\mu}_b\right)\left(G_b\left(\mathbf{x}_i\right) - \boldsymbol{\mu}_b\right)^\top, \tag{2}$$

where $N$ represents the number of samples used for fitting, and $\boldsymbol{\mu}_b$ and $\Sigma_b$ represent the estimated mean and covariance matrix for all samples used for training.

The high-level idea is to have a metric that can reflect both the sample similarity within the same label as well as sample difference across different labels. For example, an easy-classified sample should be close to the mean vector of assigned labels (representative) and far from the mean vectors of observed samples (discriminative), estimated based on the Gaussian model. Therefore, for one sample $\mathbf{x}_i$ with label $y_i = k$, we can define its RMD based on the difference of MD computed based on $G_k$ and $G_b$ as:

$$\mathcal{RMD}_k \left(\mathbf{x}_i, k\right) = \mathcal{MD}_k \left(\mathbf{x}_i, k\right) - \mathcal{MD}_b \left(\mathbf{x}_i\right), \tag{3}$$

where $\mathcal{MD}_k$ and $\mathcal{MD}_b$ represent the Mahalanobis distance computed based on samples and different clustering centroid. The formal computation of $\mathcal{MD}$ is:

$$\mathcal{MD}_k \left(\mathbf{x}_i, k\right) = -\left(G_k \left(\mathbf{x}_i\right) - \boldsymbol{\mu}_k\right)^\top \Sigma^{-1} \left(G_k \left(\mathbf{x}_i\right) - \boldsymbol{\mu}_k\right),$$
$$\mathcal{MD}_b \left(\mathbf{x}_i\right) = -\left(G_b \left(\mathbf{x}_i\right) - \boldsymbol{\mu}_b\right)^\top \Sigma_b^{-1} \left(G_b \left(\mathbf{x}_i\right) - \boldsymbol{\mu}_b\right). \tag{4}$$

However, fitting a model based on training datasets still has the risk of estimating wrong difficulty levels. For example, there exist samples with low $\mathcal{RMD}$ with misclassified results and samples with high $\mathcal{RMD}$ but correctly assigned labels based on a simple linear classifier, shown in Appendix 8.2. Moreover, the consistency of sample difficulty is also important in the estimation, and Appendix 8.2 shows that using different splits do not change the proportion of difficult samples significantly. Therefore, we estimate a prior from several fitted LR models based on the cross-validation approach. For the given training dataset $\mathcal{D}_{train}$, we split the dataset into $q$ folds based on cross-validation and fit $q$ LR models. By collecting all the samples wrongly classified by these models, we can have a list containing $n_q$ difficult samples derived from simple classifiers, denoted as $\{\mathbf{x}_i\}_{i=1}^{n_q}$, which can be further converted into the indicator weight $w$. This approach can also be used to determine whether we need to fit a neural-network-based classifier for the given problem. Moreover, we assign the maximal $\mathcal{RMD}$ for these samples, and the modified distance is defined as $\mathcal{MRMD}$ with the indicator weight. Therefore, if $w_{\mathbf{x}_i} = 1$, the $\mathcal{MRMD}$ for sample $\mathbf{x}_i$ is defined as:

$$\mathcal{MRMD}_{y_i} \left(\mathbf{x}_i, y_i\right) = \max_j \mathcal{RMD}_{y_j} \left(\mathbf{x}_j, y_j\right), \tag{5}$$

otherwise $\mathcal{MRMD}$ is the same as the pre-computed $\mathcal{RMD}$. When we train the model to classify sample $\mathbf{x}_i$, we regularize the classification loss function by treating $\mathcal{MRMD}$ as adaptive weights:

$$\mathcal{L}_f = \mathcal{L}_c \left(f_\theta \left(\mathbf{x}_i\right), y_i\right) + \alpha s \left(\mathbf{x}_i, y_i\right) \mathcal{H} \left[f_\theta \left(\mathbf{x}_i\right)\right],$$
$$s \left(\mathbf{x}_i, y_i\right) = \frac{\exp \left(\mathcal{MRMD} \left(\mathbf{x}_i, y_i\right) / T\right)}{\max_j \left\{\exp \left(\mathcal{MRMD} \left(\mathbf{x}_j, y_j\right) / T\right\} + \epsilon}, \tag{6}$$

where $L_c$ represents the cross-entropy loss, $f_\theta(\cdot)$ represents the classifier, $\mathcal{H}(\cdot)$ represents the regularized element (it can be either negative entropy or poly loss), and $s(\cdot, \cdot)$ represents the difficulty weight. $\alpha$ is the weight used for loss balancing, $T$ is the temperature parameter to control the shape of weight distribution, and $\epsilon$ represents a tiny value to avoid numerical errors. In the real application of USD to improve classification, for stable training, we normalize the distance $\mathcal{MRMD}(\mathbf{x}_i, y_i)$ into the range of $(0, 1)$. The regularized loss $\mathcal{L}_f$ can be trained with Adam Kingma (2014) optimizer. If we do not detect wrongly classified samples in this stage, our method degrades to no-prior mode. We have also provided a systemic comparison between USD and $\mathcal{RMD}$ in Appendix 8.2.

**Estimating Sample Difficulty for Regression Problems.** Previous research Pintea et al. (2023) has demonstrated that reconsidering regression problems in computer vision as classification problems can always boost model performance. Therefore, the sample difficulty of continuous labels can be estimated after transferring the continuous targets as discrete targets, for example, based on clustering methods after batch effect correction Korsunsky et al. (2019); Tran et al. (2020) or Bins-Discretizer methods Pedregosa et al. (2011). Therefore, assuming we have the transferring function $t(\cdot)$ and the discrete labels computed based on $k = t(\mathbf{y}_i)$, the difficulty of updated sample $(\mathbf{x}_i, k)$ can be defined as:

$$\mathcal{RMD}_k \left(\mathbf{x}_i, k\right) = \mathcal{MD}_k \left(\mathbf{x}_i, k\right) - \mathcal{MD}_b \left(\mathbf{x}_i\right), \tag{7}$$

where the computation of $\mathcal{MRMD}_k \left(\mathbf{x}_i, k\right)$ is the same as steps used in the classification task.

The number of clusters and bins is tuned based on maximizing the Average Silhouette Width (ASW) score Pedregosa et al. (2011). The computation process of $\mathcal{MD}_k(\cdot, \cdot)$ and $\mathcal{MD}_b(\cdot)$ is the same as the approaches used in classification. Similarly, $\mathcal{MRMD}(\cdot, \cdot)$ can also be computed based on LR with features from PFMs as inputs and discrete labels as targets.

**Learning Sample Difficulty for a General Purpose.** When considering the loss of (multi-target) regression-based problems, we propose a new correlation-aware and difficulty-aware loss function for gene expression prediction. Most of the previous work relied on minimizing the mean squared error ($\text{MSE}(\cdot, \cdot)$) of multiple genes between observed expression levels $\mathbf{y}_i$ for spot $i$ and predicted expression levels $\hat{\mathbf{y}}_i$. However, this approach only considers the global cost but ignores the fine-grained differences across spots and genes. Therefore, we first introduced the designed PCCMSE loss, which is the combination of MSE loss, spot-level Pearson correlation coefficient (PCC) loss, and gene-level PCC loss. Its definition is:

$$\mathcal{L}_{\text{base}} = \text{MSE}(\mathbf{y}, \hat{\mathbf{y}}) - \text{PCC}(\mathbf{y}, \hat{\mathbf{y}}) - \text{PCC}(\mathbf{y}^\top, \hat{\mathbf{y}}^\top). \tag{8}$$

Furthermore, inspired by Zhang et al. (2023), we also introduce the Ordinary Entropy loss function (OE) in the optimization process, which can reduce the entropy in the training process by balancing the tightness and diversity of feature space. The second term of our loss function is defined as:

$$\mathcal{L}_{\text{OE}} = -\frac{1}{M(M-1)} \sum_{i=1}^{M} \sum_{j \neq i} w_{ij} \left\| \mathbf{z}_{c_i} - \mathbf{z}_{c_j} \right\|_2 + \frac{1}{M_b} \sum_{i=1}^{M_b} \left\| \mathbf{z}_i - \mathbf{z}_{c_i} \right\|_2, \tag{9}$$

where $w_{ij} = \left\| \mathbf{y}_i - \mathbf{y}_j \right\|_2$ ensures that samples with larger distances in the expression space will receive a large penalty. Here $c_i$ and $c_j$ represent the centers in the feature space of samples $i$ and $j$, and $\mathbf{z}_i$ represents the embeddings from the outputs of the last encoder layer for the $i$-th sample. $M$ represents the number of centers and $M_b$ represents the number of samples in the given batch $b$. Finally, in our case, each feature is its center because of the expression difference, so we have $\left\| \mathbf{z}_i - \mathbf{z}_{c_i} \right\|_2 = 0$.

We finally incorporate the difficulty-aware loss function inspired by the classification problem in equation equation 6, and thus our final loss function used in USD can be represented as:

$$\mathcal{L}_{\text{final}} = \mathcal{L}_{\text{base}} + \lambda_{\text{OE}} \mathcal{L}_{\text{OE}} + \lambda_f \mathcal{L}_f, \tag{10}$$

where $\lambda_{\text{OE}}, \lambda_f$ are hyper-parameters used to control the balance of the last two loss function terms. All the hyper-parameters are tuned to the optimized version based on the model performance on the validation dataset for both baseline and proposed methods.

## 4 EXPERIMENT

### 4.1 SETUP

**Datasets.** For the disease-state classification problem, we consider three datasets covering different sub-tasks. We perform experiments of our proposed method and baseline methods for disease sub-type classification based on TCGA LUSC-LUAD (TCGA) Weinstein et al. (2013) dataset, and perform experiments for disease-state classification based on CAMELYON16 Bejnordi et al. (2017) and PANDA datasets Bulten et al. (2022). PANDA is designed as a multi-classification problem with six classes. TCGA LUSC-LUAD is a slide-level small-scale dataset and the latter two are slide-level large-scale datasets. We generate training/validation/testing samples for these three datasets randomly. Label distributions are summarized in Appendix 8.4. For the spatial transcriptomics prediction as a patch-level task, we consider eight datasets named by the source diseases/tissues (IDC, READ, PRAD, LYMPH_IDC, COAD, CCRCC, Brain, and Skin) from the HEST-1k database Jaume et al. (2024a) and STImage-1K4M database Chen et al. (2024a). The highly variable genes used for training and prediction are pre-defined in these datasets. Each dataset corresponds to one cancer or tissue type, and we filter the disease dataset whose number of batches is lower than three, which is the minimal number we need to split the whole dataset into training/validation/testing samples.

**Evaluations.** For the classification task, we select metrics Pedregosa et al. (2011) including Accuracy (Acc), Balanced Accuracy (Bacc), Kappa coefficient (Kappa), Weighted-F1 score (wF1), Area Under the Receiver Operating Characteristic curve score (AUROC), and Expected Calibration Error (ECE) Kuleshov and Liang (2015). The higher the better for all metrics except ECE. Lower ECE represents better calibrating confidence. We did not include AUROC for evaluating the multi-class classification problem. For the regression task, we select metrics including spot-level PCC (SPCC), gene-level Pearson Correlation Coefficients (GPCC), and Mean Squared Error (MSE). The higher the better for

all metrics except MSE. All metrics are widely used in the related work Chen et al. (2024b); Jia et al. (2024); Liu et al. (2025a) of classification and regression tasks.

**Baseline Models.** We have considered base models including UNI v1 Chen et al. (2024b), UNI v2 Chen et al. (2024b), GigaPath Xu et al. (2024a), and ResNet 50 He et al. (2016) for generating image features. Our selection criteria are based on the related benchmarking analyses in this task Jaume et al. (2024a); Lee et al. (2024a); Zhang et al. (2025); Vaidya et al. (2025), and training strategies are inherited from Cui et al. (2023). We exclude image-text-based PFMs to avoid data leakage. For disease-state classification, we consider LS Müller et al. (2019), $L_1$ Joo and Chung (2021), Focal Mukhoti et al. (2020), Poly Leng et al. (2022), ER Pereyra et al. (2017), CE Mannor et al. (2005), WER Cui et al. (2023), and WPoly Cui et al. (2023) as baseline models, which are widely used in related work. For gene expression prediction, we consider MSE Loss Wang and Bovik (2009), Huber Loss Huber (1992), and PCCMSE Loss as baseline models. Here MSE Loss is the most frequently used loss function in this task. Details of baselines can be found in Appendix 8.5.

**Implementation Details.** We implement our method using a single H200 NVIDIA GPU and adopt mini-batch Adam training with a batch size proportion to data scale (32 for the dataset with $n_{\text{samples}} < 1000$ and 512 for the dataset with $n_{\text{samples}} > 1000$), and the batch size is also determined under the consideration of the GPU memory usage. We utilize PyTorch-lightning Falcon (2019) to train the model and evaluate different baselines accordingly. All the spatial transcriptomic data are normalized by standard pipeline from Scanpy Wolf et al. (2018). For tuning other hyper-parameters, please refer Appendix 8.6. For running time and memory usage, please refer Appendix 8.7.

| Datasets | Metrics | Base | Methods | | | | | | | | | | Best Method |
|---|---|---|---|---|---|---|---|---|---|---|---|---|---|
| | | | LS | $L_1$ | Focal | Poly | ER | CE | WER | WPoly | USD (ER) | USD (Poly) | |
| TCGA | ACC (↑) | UNI v1 | 0.420 | 0.923 | 0.913 | 0.927 | **0.933** | 0.913 | **0.933** | 0.923 | **0.933** | 0.923 | UNIv2+USD (ER) |
| | | UNI v2 | 0.520 | 0.960 | 0.933 | 0.930 | 0.963 | 0.923 | **1.000** | 0.920 | **1.000** | 0.920 | |
| | | GigaPath | 0.510 | 0.647 | 0.603 | 0.760 | 0.697 | 0.737 | 0.677 | **0.767** | 0.677 | **0.767** | |
| | | ResNet 50 | 0.517 | 0.517 | 0.653 | 0.637 | 0.520 | 0.643 | 0.640 | **0.690** | 0.627 | 0.680 | |
| | AUROC (↑) | UNI v1 | 0.446 | 0.986 | 0.987 | 0.986 | **0.994** | 0.986 | 0.994 | 0.989 | **0.994** | 0.989 | UNIv2+USD (ER) |
| | | UNI v2 | 0.543 | **1.000** | **1.000** | **1.000** | 0.997 | **1.000** | **1.000** | **1.000** | **1.000** | **1.000** | |
| | | GigaPath | 0.570 | 0.742 | 0.838 | 0.859 | 0.877 | 0.866 | **0.898** | 0.891 | **0.898** | 0.891 | |
| | | ResNet 50 | 0.545 | 0.517 | 0.704 | 0.697 | 0.568 | 0.696 | 0.664 | 0.721 | 0.677 | **0.731** | |
| CAMELYON16 | ACC (↑) | UNI v1 | 0.494 | 0.715 | 0.715 | 0.726 | 0.732 | 0.747 | 0.724 | 0.724 | 0.741 | **0.756** | UNIv1+USD (Poly) |
| | | UNI v2 | 0.450 | 0.574 | **0.585** | 0.559 | 0.553 | 0.559 | 0.538 | 0.518 | 0.562 | 0.550 | |
| | | GigaPath | 0.491 | 0.491 | 0.468 | 0.459 | 0.482 | **0.497** | 0.488 | 0.462 | 0.491 | 0.485 | |
| | | ResNet 50 | 0.541 | 0.535 | 0.529 | 0.524 | 0.497 | 0.521 | 0.456 | 0.535 | **0.585** | 0.553 | |
| | AUROC (↑) | UNI v1 | 0.536 | 0.828 | 0.821 | 0.832 | 0.831 | 0.829 | 0.821 | 0.820 | 0.812 | **0.834** | UNIv1+USD (Poly) |
| | | UNI v2 | 0.463 | 0.752 | 0.690 | 0.738 | 0.738 | 0.724 | 0.701 | 0.713 | **0.753** | 0.739 | |
| | | GigaPath | 0.519 | **0.649** | 0.593 | 0.619 | 0.610 | 0.610 | 0.592 | 0.575 | 0.643 | 0.661 | |
| | | ResNet 50 | 0.524 | 0.725 | 0.720 | 0.719 | 0.719 | 0.725 | 0.515 | 0.712 | **0.730** | 0.703 | |
| PANDA | ACC (↑) | UNI v1 | 0.147 | 0.489 | 0.484 | 0.490 | 0.474 | 0.471 | 0.485 | 0.485 | **0.495** | 0.494 | UNIv1+USD (ER) |
| | | UNI v2 | 0.165 | 0.479 | 0.479 | **0.489** | 0.480 | 0.479 | 0.474 | 0.485 | 0.488 | 0.468 | |
| | | GigaPath | 0.182 | 0.468 | 0.459 | 0.470 | 0.460 | 0.465 | 0.466 | 0.438 | **0.473** | 0.453 | |
| | | ResNet 50 | 0.178 | 0.417 | 0.437 | 0.440 | 0.437 | **0.439** | 0.429 | 0.438 | 0.430 | 0.431 | |
| | wF1 (↑) | UNI v1 | 0.099 | 0.458 | 0.468 | 0.467 | 0.445 | 0.446 | 0.467 | 0.469 | **0.479** | 0.478 | UNIv1+USD (ER) |
| | | UNI v2 | 0.165 | 0.479 | 0.479 | **0.489** | 0.480 | 0.479 | 0.474 | 0.485 | 0.488 | 0.468 | |
| | | GigaPath | 0.171 | 0.437 | 0.435 | 0.445 | 0.438 | 0.446 | 0.428 | 0.440 | **0.455** | 0.424 | |
| | | ResNet 50 | 0.174 | 0.342 | **0.416** | 0.413 | 0.415 | **0.416** | 0.403 | 0.413 | 0.397 | 0.404 | |

Table 1: Benchmarking results across base models and training strategies for classification tasks. We reported the average scores for each method from five random seeds, and the information on standard deviation can be found in Appendix 8.8. Our proposed method and the best score are boldfaced.

## 4.2 EXPERIMENTAL RESULTS

**Disease State Classification.** We select Acc and AUROC for evaluating the dataset with binary labels, while Acc and wF1 are presented for evaluating the dataset with multiple labels, summarized in Table 1. We also provide tables with full metrics, which are listed in Appendix 8.8. We first consider LR as a simple baseline for assessing the necessity of performing training with non-linear models based on Appendix 8.9, which shows that PFMs with USD are always better than LR across different datasets. Overall, if we consider evaluating the training strategies based on different PFMs (including 12 combinations), USD achieves the highest performance in 75.0% choices evaluated by AUROC or wF1 and 53.8% choices evaluated by Acc, demonstrating the consistent improvement of USD. Furthermore, the ER mode of USD is more helpful for handling datasets with complicated structures (e.g., multi-label classification) and can also reduce the uncertainty when making the decision, reflected by the lower ECE. If we focus on a specific dataset such as TCGA, the best combination, UNI v2 and USD with ER mode, can surpass the second-best combination by 3.8%.

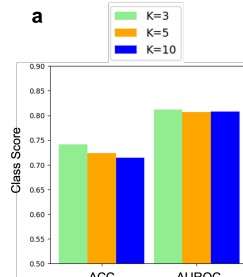 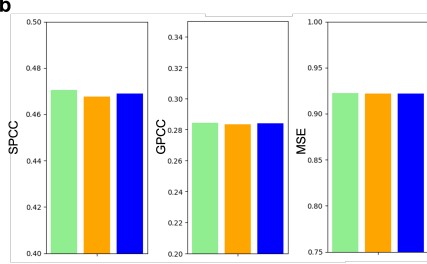

Figure 3: Results of choosing different $K$. (a) represents the results performed for the classification task. (b) represents the results performed for the regression task.

The Poly mode of USD is more suitable for datasets with simpler structures. Both of the proposed modes have low standard deviation, shown in Appendix 8.8. We also demonstrate the robustness of USD under imbalanced or noisy labels, shown in Appendix 8.4. As a result, USD acts as an efficient solution to improve the accuracy of image classification on a wide range of problem types and data, and can be easily integrated into arbitrary training pipelines for classification tasks.

| Metrics | Methods | Datasets and Rank | | | | | | | | | |
|---------|---------|------|------|------|-----------|------|------|-------|------|---------|----------|
| | | IDC | READ | PRAD | LYMPH_IDC | COAD | CCRCC | Brain | Skin | Average | Avg Rank |
| SPCC (↑) | MSE Loss | 0.581 | 0.332 | **0.691** | 0.103 | 0.621 | 0.373 | 0.602 | 0.400 | 0.463 | 2.89 |
| | Huber Loss | **0.589** | 0.322 | 0.684 | 0.090 | **0.626** | 0.369 | 0.605 | 0.393 | 0.460 | 3.22 |
| | PCCMSE Loss | 0.588 | 0.360 | 0.687 | 0.049 | 0.621 | 0.383 | 0.615 | 0.401 | 0.463 | 2.67 |
| | **USD (ER)** | **0.589** | **0.381** | 0.689 | **0.129** | 0.622 | **0.386** | **0.618** | **0.409** | **0.478** | **1.22** |
| GPCC (↑) | MSE Loss | 0.389 | 0.190 | 0.132 | 0.242 | 0.565 | 0.264 | 0.095 | 0.237 | 0.264 | 3.22 |
| | Huber Loss | 0.390 | 0.166 | 0.134 | **0.250** | **0.568** | 0.251 | 0.102 | 0.198 | 0.257 | 2.89 |
| | PCCMSE Loss | **0.400** | 0.271 | 0.134 | 0.219 | 0.562 | **0.284** | 0.133 | **0.266** | 0.284 | 2.22 |
| | **USD (ER)** | **0.400** | **0.283** | **0.138** | 0.236 | 0.565 | 0.273 | **0.154** | 0.265 | **0.289** | **1.67** |
| MSE (↓) | MSE Loss | 2.825 | 0.264 | **0.293** | **0.845** | 0.959 | 0.491 | 0.281 | 1.561 | 0.940 | 2.56 |
| | Huber Loss | 2.812 | **0.228** | 0.301 | 0.864 | 0.969 | 0.499 | **0.279** | 1.578 | 0.941 | 3.22 |
| | PCCMSE Loss | **2.748** | 0.242 | **0.293** | 0.769 | 0.958 | **0.486** | 0.285 | 1.578 | **0.920** | **1.89** |
| | **USD (ER)** | 2.754 | 0.269 | 0.294 | 0.857 | **0.957** | 0.492 | **0.279** | **1.481** | 0.923 | 2.33 |

Table 2: Benchmarking results for the regression task. We report the average scores (Average) for each method from five random seeds and average rank (Avg Rank) by averaging method's rank in different datasets. The information on standard deviation can be found in Appendix 8.8. USD and the score with best value are boldfaced, and lower rank represents a better method.

**Gene Expression Prediction.** We first select the most promising PFM to form the base model for predicting spatial transcriptomics based on PCCMSE Loss. According to the Appendix 8.8, UNI v2 is the best option for predicting gene expression levels from patches, so we conduct main experiments based on this model to reduce the cost of generating path-level embeddings for each dataset, estimating the sample difficulty, and training different models for expression prediction. According to Table 2, MSE Loss and Huber Loss generally perform worse than PCCMSE Loss, reflected in the lower SPCC score and GPCC score, as well as higher MSE, on average. USD also surpasses state-of-the-art training framework, DeepPT Hoang et al. (2024), discussed in Appendix 8.10. Moreover, USD achieves the highest SPCC score in 75% datasets and the highest GPCC score in 50% datasets. Compared with the second-best method in the selected metrics, USD makes an average improvement by 3.2% for SPCC and 1.8% for GPCC. If we compare USD with MSE Loss, which is a more generally used loss function in this task, we can improve the model performance by 62.2% at most for GPCC in the Brain dataset. USD also has low variance, validated by the table with information of the standard deviation. Therefore, USD can participially predict gene expression levels higher than the baselines based on the cross-gene evaluation setting, which is closer to the practical applications of gene expression analysis, such as the detection of differential expression gene Kiselev et al. (2019); Song et al. (2023b) and the selection of cell-type-specific marker genes Pullin and McCarthy (2024).

## 4.3 ANALYSIS

**Insights from Analyzing Factors Affecting Image Classification.** To estimate the sample difficulty with prior, we need to run $K$-fold cross-validation to collect the samples that are wrongly predicted

by a simple linear predictor. By adjusting different $K$, we have various sample lists with different lengths. To determine a suitable $K$ and demonstrate the robustness of our method, we examine different $K$ based on the CAMELYON16 dataset with base model UNI v1. According to Figure 3 (a), increasing $K$ may slightly reduce model performance, which shows that our training strategy expects a relatively smaller $K$ to generate difficult sample sets. Moreover, very large $K$ requires longer training time, and thus we finally fix $K = 3$ for all datasets. We also consider the options of input type with different modes, the necessity of dropping the difficult samples or fine-tuning the base model and prediction head together, and the options of computing sample difficulty, discussed in Appendix 8.11. These variations cannot make improvement.

**Lessons from Analyzing Factors Affecting Gene Expression Prediction.** In the regression task, based on Figure 3 (b), adjusting $K$ will not affect model performance too much, and thus USD is very robust to $K$ in the gene expression prediction setting. We have included a similar study for the cluster number with all datasets in Appendix 8.12. Furthermore, we perform ablation studies to investigate the contribution of different loss function components, summarized in Figure 4 (a). According to this figure, our final loss function $\mathcal{L}_{\text{final}}$ has the highest SPCC and GPCC scores, while its MSE is close to the best method. Moreover, we find that incorporating the term $\mathcal{L}_{\text{OE}}$ can help us better learn the cell-level and gene-level correlations while adding the term $\mathcal{L}_f$ regularized by the sample difficulty helps us reducing the average error between predicted and observed expression levels. This conclusion matches with previous studies arguing that utilizing classification loss can reduce the MSE for the regression task. If we do not consider incorporating sample difficulty and use cross entropy (CE) to compute the classifi-

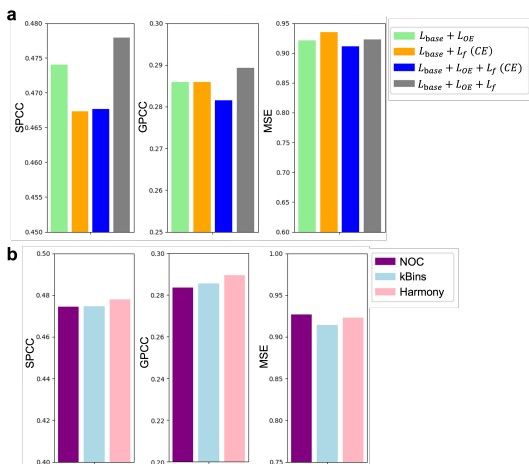

Figure 4: Ablation tests for the regression task. (a) represents the results under different components of final loss. The mode *CE* means using cross entropy as the classification loss. (b) represents the results under different batch effect correction strategies.

cation loss, we cannot achieve improvement. We also consider the approaches to reduce batch effect in the expression space, including Harmony, kBins, and no correction mode (NoC), and the results are summarized in Figure 4 (b). Correcting batch effect can improve model performance. Running Harmony or KBins can make the correlation smoother and reduce the batch effect in the relationship of SPCC and difficulty, shown in Figure 5. The comparisons of different modes and base models are summarized in Appendices 8.13 and 8.14.

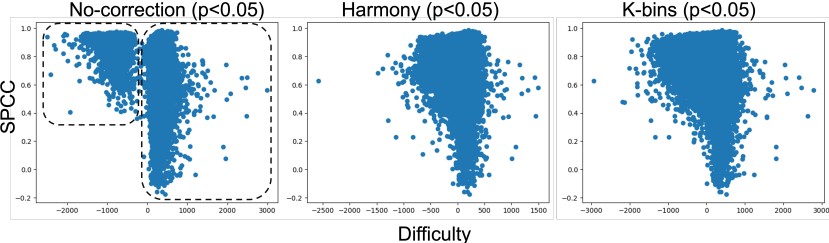

Figure 5: Relationship between sample difficulty and SPCC based on adjusting different batch effect correction strategies. The SPCC is computed based on the training dataset.

## 5 CONCLUSION

This paper investigates a clinical-associated problem of estimating the slide-level or patch-level training difficulty to boost model performances targeting two typical tasks in histopathology image analysis, including the classification of disease states and the prediction of spatial transcriptomics. We have also included a section in Appendix 8.15 to discuss limitations.

## 6 ETHICS STATEMENT

All authors follow the ethics statement of this conference. The users are solely responsible for the content they generate with models in USD, and there are no mechanisms in place for addressing harmful, unfaithful, biased, and toxic content disclosure. Any modifications of the models should be released under different version numbers to keep track of the original models related to this manuscript. The target of current USD only serves for academic research. The users cannot use it for other purposes.

## 7 REPRODUCIBILITY STATEMENT

We have provided source codes in the abstract and supplementary files for reproductibility. We have also provided detailed scores of all methods tested in our submission.

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

# 8 APPENDIX

In this section, we present information on baselines, hyper-parameters, and other analyses or tables that cannot be placed in the main text due to page limitation.

## 8.1 VISUALIZATION OF SAMPLE DIFFICULTY.

Here we visualize the sample label as well as sample difficulty based on the TCGA dataset with UNI v1 embeddings Figure 6 based on UMAP McInnes et al. (2018). According to this figure, we capture sample difficulty of different labels, and the samples with similar difficulty levels show clustering performances. This discovery further conforms our interpretation of sample difficulty.

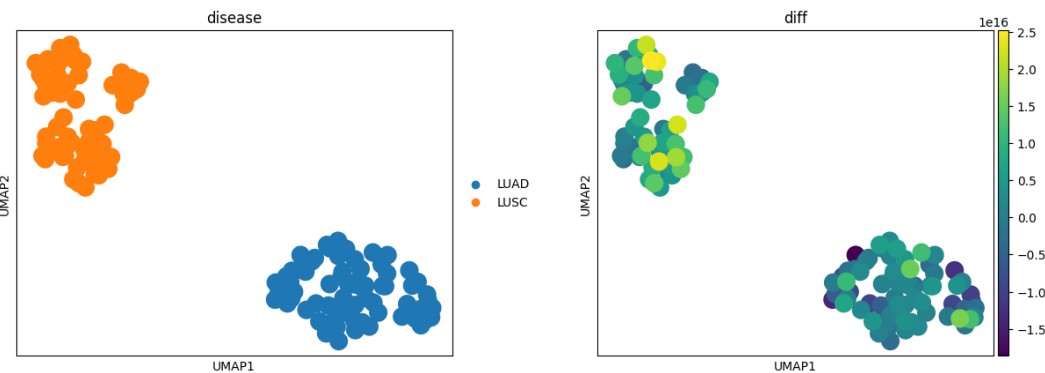

Figure 6: UMAP visualization of sample embeddings colored by disease states (left) and sample difficulty (right).

## 8.2 MOTIVATIONS AND STABILITY OF *MRMD*.

**Motivation explanation.**

According to Figure 7, we found that Logistic Regression (LR) can make correct prediction for samples with high difficulty levels, as well as wrong prediction for samples with low difficulty levels. This observation motivates us to reconsider the design of sample difficulty estimation, as we need to include the prior from a simple regression before estimating the sample difficulty with a more complicated model. Since the main purpose of considering sample difficulty is to improve generalizability by correctly predicting difficult samples, we believe it is necessary to reconsider the definition of difficult samples.

**Stability explanation.**

We ensure the consistency of sample difficulty by examining the consistency of the proportion of wrongly classified labels with different numbers of cross-validation sets. Here we show the proportion overlap by iterating different split $q$ in Tables 3 and 4, and we do not observe strong oscillation by iterating different $q$ for the three datasets used in the classification task. Therefore, our proposed method can define a robust method for generating difficult samples.

| **Number of split** | 2 | 3 | 4 | 5 | 6 | 7 | 8 | 9 | 10 |
|---|---|---|---|---|---|---|---|---|---|
| Proportion | 0.46 | 0.43 | 0.49 | 0.49 | 0.47 | 0.44 | 0.44 | 0.43 | 0.43 |

Table 3: Relationship between the number of splits and the proportion of difficult samples identified by LR model in the CAMELYON16 dataset.

## 8.3 COMPARISON BETWEEN RMD AND USD.

- Different scenarios: Cui et al. (2023) focuses on a general computer vision problem with public datasets from different domains, but USD aims to tackle a challenge mentioned in

Difficulty: <<0
Predicted: LUAD
GT Label: LUSC

Difficulty: >>0
Predicted: LUAD
GT Label: LUAD

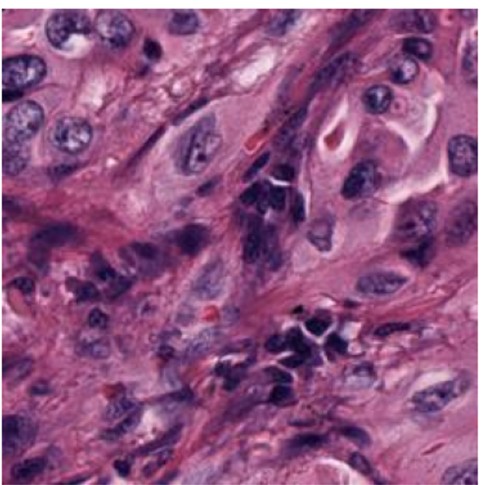
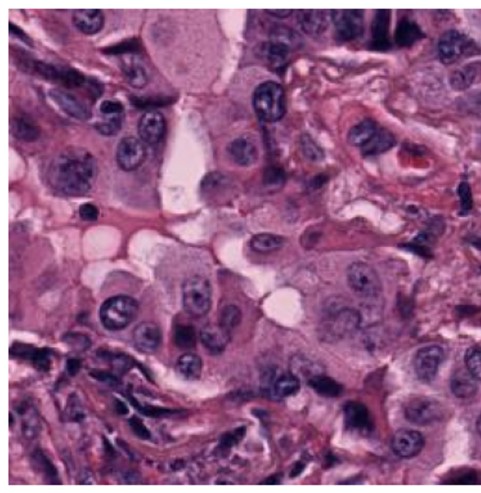

Figure 7: Examples of histopathology images and corresponding decisions made by LR, which is based on ResNet 50 and TCGA dataset.

| Number of split | 2 | 3 | 4 | 5 | 6 | 7 | 8 | 9 | 10 |
|---|---|---|---|---|---|---|---|---|---|
| Proportion | 0.59 | 0.57 | 0.56 | 0.56 | 0.55 | 0.56 | 0.55 | 0.55 | 0.55 |

Table 4: Relationship between the number of splits and the proportion of difficult samples identified by LR model in the PANDA dataset.

the potential limitations of Cui et al. (2023), which focuses on medical image analysis, as the medical images suffer from more challenging scenarios, such as label imbalance and noisy data. Cui et al. (2023) is not straightforwardly suitable for medical domain data.

- Different problem settings: In terms of problem construction, Cui et al. (2023) only considers image classification as a major task, while USD considers more diverse tasks, including image classification as well as gene expression prediction (a regression task). We are among the first research groups that try to improve the image regression prediction performance by leveraging the estimated sample difficulty, and thus USD is a more generalizable tool. Moreover, extending the estimation method to regression problems is not easily shown in our comprehensive experiments and discussion.

- Different difficult estimation methods: In terms of the estimation of sample difficulty, Cui et al. (2023) does not consider any prior information which might help on the estimation process, while USD considers using a simpler classifier such as Logistic Regression to provide correct prior to estimate a more accurate sample difficulty, supported by the visualization result in Figure 1 and the performance improvement across different tasks.

- Different experimental designs: Cui et al. (2023) did not consider many ablation studies and did not justify the necessity of introducing sample difficulty estimation for different datasets, as it lacked comparison with a linear-based classifier, but USD introduces a more rigorous comparison and demonstrates that we need to figure out the complexity of the problem ahead of model training and construction. USD also has more ablation studies to justify our choices for both the classification and regression tasks. Therefore, USD improves significantly and generalizes to a different area compared with Cui et al. (2023), which leads to an independent method.

## 8.4 LABEL DISTRIBUTION OF IMBALANCE AND NOISY DATA TESTING.

**Label distributions of datasets designed for the classification task.**

| Type | TCGA_number |
|------|-------------|
| tumor | 100 |
| health | 90 |

Table 5: Label distribution of the TCGA dataset.

| Type | CAMELYON16_number |
|------|-------------------|
| tumor | 159 |
| health | 111 |

Table 6: Label distribution of the CAMELYON16 dataset.

| Index | PANDA_number |
|-------|--------------|
| 0 | 2892 |
| 1 | 2666 |
| 2 | 1343 |
| 4 | 1249 |
| 3 | 1242 |
| 5 | 1224 |

Table 7: Label distribution of the PANDA dataset.

**Analysis of label imbalance testing.**

To produce data with imbalanced labels, we now include more experiment results based on sampling the labels to create an extreme imbalance dataset from CAMELYON16, shown in Table 8, including the case of many positive samples and the case of many negative samples. According to the results from this table, USD still performs well under the extreme conditions, achieving over 80% accuracy under two situations. Compared with the original dataset, we even have better performance, and thus USD will not be affected by the issue of label imbalance significantly.

| Dataset | Model | Metric | Original | Many positive samples (pos/neg=7) | Many negative samples (neg/pos=14) |
|---------|-------|--------|----------|-----------------------------------|------------------------------------|
| CAMELYON16 | UNI v1+USD (Poly) | ACC | 0.756 (0.02) | 0.850 (0.03) | 0.949 (0.01) |
| | | wF1 | 0.746 (0.03) | 0.804 (0.01) | 0.804 (0.01) |
| | UNI v1+USD (ER) | ACC | 0.741 (0.03) | 0.831 (0.03) | 0.977 (0.00) |
| | | wF1 | 0.750 (0.03) | 0.988 (0.00) | 0.766 (0.01) |

Table 8: Performances of USD with two different sample difficulty penalty methods under three conditions with label imbalance simulation.

**Analysis of label noise testing.**

To produce data with noisy labels, we utilize the symmetric noise generation method used in trustworthy machine learning Zhang et al. (2024) and discrete diffusion models Lou et al. (2023), which means we select a certain proportion of samples and randomly pick different labels to replace their correct labels. We then train USD with pathology image features from UNI v1 for the classification task. According to Table 9 with Accuracy, AUROC, and wF1 metrics, we find that USD still shows good performances under the condition with relatively lower noise label proportion (0.1-0.3), and the performance USD will be affected under high noise level proportion, which aligns with the study of label noise shown in Zhang et al. (2024). Therefore, USD is still a robust mode for datasets with a small amount of imperfect labels. In real applications, for datasets with a very high proportion of imperfect labels, which might be caused by low data quality and the calibration with domain experts, label re-annotation, loss re-design could be more suitable approaches in medical applications Shi et al. (2024).

## 8.5 EXPLANATIONS OF BASELINES

**Explanations of Baseline Methods for Disease-State Classification.**

| Dataset | Metric | 0.1 | 0.3 | 0.5 | 0.7 |
|---------|--------|-----|-----|-----|-----|
| TCGA | ACC | 0.910 (0.05) | 0.923 (0.05) | 0.920 (0.03) | 0.090 (0.04) |
|      | AUROC | 0.991 (0.01) | 0.992 (0.01) | 0.990 (0.01) | 0.016 (0.02) |
| CAMELYON16 | ACC | 0.665 (0.02) | 0.594 (0.04) | 0.526 (0.04) | 0.365 (0.04) |
|      | AUROC | 0.798 (0.02) | 0.671 (0.01) | 0.491 (0.04) | 0.319 (0.03) |
| PANDA | ACC | 0.461 (0.01) | 0.429 (0.00) | 0.389 (0.01) | 0.295 (0.04) |
|      | wF1 | 0.438 (0.01) | 0.396 (0.01) | 0.340 (0.03) | 0.217 (0.07) |

Table 9: Model performances with the format score (standard deviation) under different noise levels (0.1–0.7) for the classification task.

Let $p_k$ be the likelihood that the model assigned to the $k$-th class given the input $\mathbf{x}$, and $y_k$ is the true target, where $y_k$ is 1 for the correct class and 0 for the rest.

- Cross Entropy (CE): $\mathcal{L}_{\text{CE}} = -\sum_{k=1}^{K} y_k \log p_k$.

- Label Smoothing (LS): $\mathcal{L}_{\text{LS}} = -\sum_{k=1}^{K} y_k^{\text{LS}} \log p_k$ with $y_k^{\text{LS}} = y_k(1-\alpha) + \alpha/K$ and $\alpha$ is a tuning parameter.

- Focal Loss: $\mathcal{L}_{\text{Focal}} = -\sum_{k=1}^{K} y_k(1-p_k)^{\gamma} \log p_k$, where $\gamma$ is a tuning parameter.

- Entropy Regularizer (ER): $\mathcal{L}_{\text{ER}} = \mathcal{L}_{\text{CE}} - \alpha \mathcal{H}(p)$, where $\mathcal{H}(p) = -\sum_{k=1}^{K} p_k \log p_k$ and $\alpha$ is a tuning parameter.

- Poly-N Loss: $\mathcal{L}_{\text{Poly}} = \mathcal{L}_{\text{CE}} + \sum_{k=1}^{K} y_k \sum_{j=1}^{N} \epsilon_j (1-p_k)^j$ where $\epsilon_j$ is the perturbation term for the $j$-th coefficient.

- $L_1$ Loss: $\mathcal{L}_{L_1} = \mathcal{L}_{\text{CE}} + \lambda \|f_W\|_1$ where $f_W \in \mathbb{R}^K$ is the logit values, and we use it to compute $p_k = \text{softmax}_k(f_W)$.

- Weighted ER: $\mathcal{L}_{\text{WER}} = \mathcal{L}_{\text{CE}} - \alpha s(\mathbf{x}, y) \mathcal{H}(p)$, where $\alpha$ is a tuning parameter and $s(\mathbf{x}, y)$ is a sample-specific weighting derived from the RMD-based sample difficulty score.

- Weighted Poly-1:

$$\mathcal{L}_{\text{WPoly}} = \mathcal{L}_{\text{CE}} + s(\mathbf{x}, y) \sum_{k=1}^{K} y_k \epsilon_1 (1-p_k),$$

  where $s(\mathbf{x}, y)$ is a sample-specific weighting derived from the RMD-based sample difficulty score.

**Explanations of Baseline Methods for Gene Expression Prediction.**

Let $y$ be the true target and $f(\mathbf{x})$ be the prediction based on the input $\mathbf{x}$.

- MSE Loss: $\mathcal{L}_{\text{MSE}} = (y - f(\mathbf{x}))^2$.

- Huber Loss: Given a hyper-parameter $\delta$,

$$\mathcal{L}_{\text{Huber}} = \begin{cases} \frac{1}{2}(y - f(\mathbf{x}))^2 & \text{if } |y - f(\mathbf{x})| \leq \delta \\ \delta(|y - f(\mathbf{x})| - \frac{1}{2}\delta) & \text{otheriwse} \end{cases}.$$

**Explanations of Methods for Reducing Batch Effect.** Batch effect means the technique noise existing in the sequencing data from different samples. We consider Harmony Korsunsky et al. (2019) and KBins Pedregosa et al. (2011) as two approaches for reducing batch effect. The idea of Harmony is to utilize iterative clustering to pull the cells (spots) from different samples with similar biological information to a cluster, until the convergence. This approach has been validated by several benchmarking studies Tran et al. (2020); Arevalo et al. (2024) as a suitable method. KBins means we utilize k-bin discreter to place spots with similar average gene expression profiles across genes in a cluster, and thus the batch effect can be reduced by better characterizing biology-informed clusters.

## 8.6 HYPER-PARAMETER TUNING

For the disease-state classification task, we inherit the loss-specific hyper-parameter from Cui et al. (2023), which is already tuned. These parameters include the entropy weight $\lambda_e = 0.3$, the Focal

weight $f_\gamma = 1.0$, the LS weight $\epsilon = 1.0$, the $L_1$ weight $\alpha = 1.0$, and the Poly weight $\epsilon_p = 2.0$. The learning rate for training different combinations with PANDA and CAMELYON16 is 1e-3. The learning rate for training different combinations based on UNI v1, UNI v2, and GigaPath is 1e-3, and 1e-2 based on ResNet 50, for the TCGA dataset. The choice of fold $s$ is explained in the Analysis section. In this section, we present information on baselines, hyper-parameters, and other analyses or tables that cannot be placed in the main text due to page limitations.

For the gene expression prediction task, we tune the learning rate, $\lambda_{\text{OE}}$, and $\lambda_f$ based on the grid search for all models. The final choices of these three parameters are summarized in Table 10. We found that the change of these choices is not in a large range, and thus our model is robust for different conditions. The choice of fold $s$ is explained in the Analysis section.

| Dataset | Learning Rate | $\lambda_{\text{OE}}$ | $\lambda_f$ |
|---|---|---|---|
| IDC | 1.00E-04 | 1.00E-03 | 1.00E-03 |
| READ | 1.00E-03 | 1.00E-03 | 1.00E-03 |
| PRAD | 1.00E-03 | 1.00E-03 | 1.00E-03 |
| LYMPH_IDC | 1.00E-03 | 1.00E-03 | 1.00E-02 |
| COAD | 1.00E-03 | 1.00E-03 | 1.00E-03 |
| CCRCC | 1.00E-03 | 1.00E-03 | 1.00E-03 |
| Brain | 1.00E-03 | 1.00E-03 | 1.00E-03 |
| Skin | 1.00E-03 | 1.00E-03 | 1.00E-03 |

Table 10: Hyper-parameter tuning information of the spatial transcriptomic prediction task.

## 8.7 TRAINING EFFICIENCY

Here we present the running time and consumed GPU memory in Table 11 for the classification task and Table 12 for the regression task. According to these tables, USD consumes comparable resources with other baselines, but can improve model performances.

| Method | Time (s) | GPU memory usage (GB) |
|---|---|---|
| LS | 68.457 | 4.725 |
| $L_1$ | 73.973 | 4.725 |
| Focal | 60.086 | 4.725 |
| Poly | 68.852 | 4.725 |
| ER | 77.329 | 4.266 |
| CE | 79.456 | 4.725 |
| WER | 111.930 | 4.396 |
| Wpoly | 108.710 | 4.396 |
| USD (Poly) | 108.710 | 4.396 |

Table 11: Running time and memory usage for the classification task. We include statistics from both baseline methods and USD. The experiment is performed on the CAMELYON16 dataset.

| Method | Time (s) | GPU memory usage (GB) |
|---|---|---|
| MSE Loss | 307.252 | 5.930 |
| Huber Loss | 306.928 | 5.930 |
| PCCMSE Loss | 335.070 | 5.930 |
| USD (ER) | 931.392 | 11.129 |

Table 12: Running time and memory usage for the classification task. We include statistics from both baseline methods and USD. The experiment is performed on the Brain dataset.

## 8.8 FULL TABLES

We list the average scores of all metrics for the classification task in Table 13, the standard deviation of all metrics for the classification task in Table 14, and the standard deviation of all metrics for the regression task in Table 15.

## 8.9 COMPARISONS BETWEEN LOGISTIC REGRESSION AND USD FOR DISEASE-STATE PREDICTION.

Here we consider a simple baseline, Logistic Regression (LR), and fit this model then make a comparison with our proposed model, to demonstrate the necessity of using the more advanced approach to address the disease-state classification task. According to Table 16, our proposed method performs better than LR in all of the included metrics across three datasets, and thus we demonstrate the necessity of developing a novel solution for this task.

## 8.10 COMPARISONS BETWEEN USD AND TASK-SPECIFIC METHOD DEEPPT FOR GENE EXPRESSION PREDICTION.

Here we include the comparison between our proposed method and a task-specific method DeepPT, which was benchmarked in a recent publication for gene expression prediction from histopathology images and ranked as the best method Zhang et al. (2025). DeepPT encodes a patch into embedding space with pre-trained models, and later trains an auto-encoder to make the image embeddings become more dense, and the compressed embeddings are used for predicting gene expression levels. According to Table 17, USD performs better than DeepPT evaluated by all metrics on average, and participially in the READ and the LYMPH_IDC datasets. Table 18 shows that USD is also a robust method with low variance. Therefore, USD can also surpass current state-of-the-art training pipeline.

## 8.11 COMPARISONS OF METHODS FOR TRAINING THE PREDICTION HEAD

Here we consider two modes of training the prediction head for disease-state classification based on the TCGA dataset. The first mode is full parameter training (FPT), which means we tune the feature extractor together with the prediction head. The second mode is only training the prediction head (TPH) and freezing the feature extractor, which is also the default mode with less GPU memory usage. According to Table 19, TPH performs better than FPT in all metrics, and thus we keep TPH as our final solution.

We also investigate the contribution of using patch-level (36 patches per image) information from the whole slide to train a classifier for disease-state prediction with mean pooling (MP) and multi-instance learning (ABMIL). The comparison based on the CAMELYON16 dataset with UNI v1 as base model is shown in Table 20. According to this table, using PFMs to encode slides directly is a better choice, and its required scale of training data is smaller than multi-instance learning design. The potential limitations of patch-based methods such as MP and ABMIL Ilse et al. (2018) are the bias in selecting patches to represent a slide, and the training cost of patch-level information is also more expensive. Nevertheless, our conclusion in the slide-level representation can also be transferred to patch-level representation easily, demonstrated by our regression-based experiments. Moreover, we consider removing samples which are wrongly classified by the linear classifier and re-train the prediction head (RD), whose result is also summarized in this table. We find that removing difficult samples cannot improve model performance, and thus our default setting is the most optimal setting. With the same dataset, we also consider a different approach to compute $\mathcal{MRMD}$, that is, for a sample with class $c$, we compute the base Gaussian model $G_b$ based on the samples not belonging to this class. This approach is represented as $\mathcal{MRMD}$ (class removal) and the default method is represented as $\mathcal{MRMD}$ (base). According to Table x, $\mathcal{MRMD}$ (base) has better performances, and thus using different types of samples to compute $\mathcal{MRMD}$ also does not improve the performance of USD.

## 8.12 EFFECT OF CHOOSING THE CLUSTER NUMBER

Since the ASW score is widely used in evaluating the clustering performance in spatial transcriptomic data analysis, we believe that the biological signals will not be oversimplified by selecting the optimal bin number. We present additional experimental results by using different numbers of bins for

| Datasets | Metrics | Base | LS | $L_1$ | Focal | Poly | ER | CE | WER | WPoly | USD (ER) | USD (Poly) | Best Method |
|---|---|---|---|---|---|---|---|---|---|---|---|---|---|
| | | | | | | | | | | | | | Methods |
| TCGA | ACC (↑) | UNI v1 | 0.420 | 0.923 | 0.913 | 0.927 | **0.933** | 0.913 | **0.933** | 0.923 | **0.933** | 0.923 | |
| | | UNI v2 | 0.520 | 0.960 | 0.933 | 0.930 | 0.963 | 0.923 | **1.000** | 0.920 | **1.000** | 0.920 | UNI v2+USD (ER) |
| | | GigaPath | 0.510 | 0.647 | 0.603 | 0.760 | 0.697 | 0.737 | 0.677 | **0.767** | 0.677 | **0.767** | |
| | | ResNet 50 | 0.517 | 0.517 | 0.653 | 0.637 | 0.520 | 0.643 | 0.640 | **0.690** | 0.627 | 0.680 | |
| | AUROC (↑) | UNI v1 | 0.446 | 0.986 | 0.987 | 0.986 | **0.994** | 0.986 | 0.994 | 0.989 | **0.994** | 0.989 | |
| | | UNI v2 | 0.543 | **1.000** | **1.000** | **1.000** | 0.997 | **1.000** | **1.000** | **1.000** | **1.000** | **1.000** | UNI v2+USD (ER) |
| | | GigaPath | 0.570 | 0.742 | 0.838 | 0.859 | 0.877 | 0.866 | **0.898** | 0.891 | **0.898** | 0.891 | |
| | | ResNet 50 | 0.545 | 0.517 | 0.704 | 0.697 | 0.568 | 0.696 | 0.664 | 0.721 | 0.677 | **0.731** | |
| | Bacc (↑) | UNI v1 | 0.420 | 0.923 | 0.913 | 0.927 | **0.933** | 0.913 | **0.933** | 0.923 | **0.933** | 0.923 | |
| | | UNI v2 | 0.520 | 0.960 | 0.933 | 0.930 | 0.963 | 0.923 | **1.000** | 0.920 | **1.000** | 0.920 | |
| | | GigaPath | 0.510 | 0.647 | 0.603 | 0.760 | 0.697 | 0.737 | 0.677 | **0.767** | 0.677 | **0.767** | |
| | | ResNet 50 | 0.517 | 0.517 | 0.653 | 0.637 | 0.520 | 0.643 | 0.640 | **0.690** | 0.627 | 0.680 | UNI v2+USD (ER) |
| | Kappa (↑) | UNI v1 | -0.160 | 0.847 | 0.827 | 0.853 | **0.867** | 0.827 | 0.867 | **0.847** | **0.867** | 0.847 | |
| | | UNI v2 | 0.040 | 0.920 | 0.867 | 0.860 | 0.927 | 0.847 | **1.000** | 0.840 | **1.000** | 0.840 | |
| | | GigaPath | 0.020 | 0.293 | 0.207 | 0.520 | 0.393 | 0.473 | 0.353 | **0.533** | 0.353 | **0.533** | |
| | | ResNet 50 | 0.033 | 0.033 | 0.307 | 0.273 | 0.040 | 0.287 | 0.280 | **0.380** | 0.253 | 0.360 | UNI v2+USD (ER) |
| | wF1 (↑) | UNI v1 | 0.361 | 0.923 | 0.913 | 0.926 | **0.933** | 0.913 | **0.933** | 0.923 | **0.933** | 0.923 | |
| | | UNI v2 | 0.467 | 0.960 | 0.933 | 0.930 | 0.963 | 0.923 | **1.000** | 0.919 | **1.000** | 0.919 | |
| | | GigaPath | 0.447 | 0.636 | 0.532 | 0.749 | 0.639 | 0.721 | 0.627 | **0.758** | 0.627 | **0.758** | |
| | | ResNet 50 | 0.488 | 0.442 | 0.651 | 0.631 | 0.438 | 0.636 | 0.629 | **0.690** | 0.613 | 0.676 | UNI v2+USD (ER) |
| | ECE (↓) | UNI v1 | 0.107 | 0.242 | 0.100 | 0.089 | **0.051** | 0.098 | 0.065 | 0.097 | 0.065 | 0.097 | |
| | | UNI v2 | 0.186 | 0.299 | 0.073 | **0.069** | 0.243 | 0.078 | 0.151 | 0.073 | 0.151 | 0.073 | |
| | | GigaPath | 0.131 | 0.166 | 0.253 | **0.121** | 0.227 | 0.137 | 0.208 | 0.126 | 0.208 | 0.126 | |
| | | ResNet 50 | 0.210 | 0.199 | **0.155** | 0.210 | 0.219 | 0.219 | 0.184 | 0.209 | 0.184 | 0.209 | UNI v1+ER |
| CAMELYON16 | ACC (↑) | UNI v1 | 0.494 | 0.715 | 0.715 | 0.726 | 0.732 | 0.747 | 0.724 | 0.724 | 0.741 | **0.756** | |
| | | UNI v2 | 0.450 | 0.574 | **0.585** | 0.559 | 0.553 | 0.559 | 0.538 | 0.518 | 0.562 | 0.550 | UNI v1+USD (Poly) |
| | | GigaPath | 0.491 | 0.491 | 0.468 | 0.459 | 0.482 | **0.497** | 0.488 | 0.462 | 0.491 | 0.485 | |
| | | ResNet 50 | 0.541 | 0.535 | 0.529 | 0.524 | 0.497 | 0.521 | 0.456 | 0.535 | **0.585** | 0.553 | |
| | AUROC (↑) | UNI v1 | 0.536 | 0.828 | 0.821 | 0.832 | 0.831 | 0.829 | 0.821 | 0.820 | 0.812 | **0.834** | |
| | | UNI v2 | 0.463 | 0.752 | 0.690 | 0.738 | 0.738 | 0.724 | 0.701 | 0.713 | **0.753** | 0.739 | UNI v1+USD (Poly) |
| | | GigaPath | 0.519 | **0.649** | 0.593 | 0.619 | 0.610 | 0.610 | 0.592 | 0.575 | 0.643 | 0.661 | |
| | | ResNet 50 | 0.524 | 0.725 | 0.720 | 0.719 | 0.719 | 0.725 | 0.515 | 0.712 | **0.730** | 0.703 | |
| | Bacc (↑) | UNI v1 | 0.491 | 0.734 | 0.733 | 0.746 | 0.752 | 0.765 | 0.741 | 0.738 | 0.757 | **0.771** | |
| | | UNI v2 | 0.456 | 0.605 | 0.615 | 0.592 | 0.587 | **0.593** | 0.573 | 0.553 | 0.592 | 0.586 | |
| | | GigaPath | 0.516 | 0.527 | 0.510 | 0.503 | 0.515 | **0.536** | 0.525 | 0.505 | 0.531 | 0.526 | |
| | | ResNet 50 | 0.539 | 0.571 | 0.565 | 0.562 | 0.536 | 0.558 | 0.500 | 0.572 | **0.617** | 0.588 | UNI v1+USD (WPoly) |
| | Kappa (↑) | UNI v1 | -0.017 | 0.449 | 0.448 | 0.472 | 0.484 | 0.510 | 0.463 | 0.460 | 0.496 | **0.525** | |
| | | UNI v2 | -0.087 | 0.197 | **0.218** | 0.173 | 0.163 | 0.174 | 0.137 | 0.100 | 0.173 | 0.160 | |
| | | GigaPath | 0.030 | 0.050 | 0.018 | 0.005 | 0.029 | **0.067** | 0.047 | 0.010 | 0.060 | 0.049 | |
| | | ResNet 50 | 0.077 | 0.133 | 0.122 | 0.114 | 0.068 | 0.108 | 0.000 | 0.135 | **0.219** | 0.165 | UNI v1+USD (WPoly) |
| | wF1 (↑) | UNI v1 | 0.494 | 0.702 | 0.704 | 0.715 | 0.721 | 0.739 | 0.715 | 0.718 | **0.750** | 0.746 | |
| | | UNI v2 | 0.435 | 0.510 | **0.521** | 0.482 | 0.475 | 0.479 | 0.445 | 0.407 | 0.498 | 0.463 | |
| | | GigaPath | **0.413** | 0.385 | 0.316 | 0.292 | 0.378 | 0.378 | 0.366 | 0.298 | 0.350 | 0.342 | |
| | | ResNet 50 | **0.534** | 0.446 | 0.438 | 0.417 | 0.371 | 0.417 | 0.286 | 0.438 | 0.524 | 0.467 | UNI v1+USD (ER) |
| | ECE (↓) | UNI v1 | **0.066** | 0.102 | 0.100 | 0.139 | 0.092 | 0.110 | 0.080 | 0.133 | 0.107 | 0.106 | |
| | | UNI v2 | **0.058** | 0.114 | 0.103 | 0.199 | 0.119 | 0.144 | 0.135 | 0.229 | 0.159 | 0.156 | |
| | | GigaPath | **0.070** | 0.142 | 0.163 | 0.342 | 0.164 | 0.194 | 0.188 | 0.324 | 0.199 | 0.232 | |
| | | ResNet 50 | **0.049** | 0.100 | 0.106 | 0.245 | 0.137 | 0.181 | 0.190 | 0.193 | 0.137 | 0.167 | ResNet 50+LS |
| PANDA | ACC (↑) | UNI v1 | 0.147 | 0.489 | 0.484 | 0.490 | 0.474 | 0.471 | 0.485 | 0.485 | **0.495** | 0.494 | |
| | | UNI v2 | 0.165 | 0.479 | 0.479 | **0.489** | 0.480 | 0.479 | 0.474 | 0.485 | 0.488 | 0.468 | UNI v1+USD (ER) |
| | | GigaPath | 0.182 | 0.468 | 0.459 | 0.470 | 0.460 | 0.465 | 0.460 | 0.466 | **0.473** | 0.453 | |
| | | ResNet 50 | 0.178 | 0.417 | 0.437 | 0.440 | 0.437 | **0.439** | 0.429 | 0.438 | 0.430 | 0.431 | |
| | wF1 (↑) | UNI v1 | 0.099 | 0.458 | 0.468 | 0.467 | 0.445 | 0.446 | 0.467 | 0.469 | **0.479** | 0.478 | |
| | | UNI v2 | 0.165 | 0.479 | 0.479 | **0.489** | 0.480 | 0.479 | 0.474 | 0.485 | 0.488 | 0.468 | UNI v1+USD (ER) |
| | | GigaPath | 0.171 | 0.437 | 0.435 | 0.445 | 0.438 | 0.446 | 0.428 | 0.440 | **0.455** | 0.424 | |
| | | ResNet 50 | 0.174 | 0.342 | **0.416** | 0.413 | 0.415 | 0.416 | 0.403 | 0.413 | 0.397 | 0.404 | |
| | Bacc (↑) | UNI v1 | 0.162 | 0.422 | 0.432 | 0.431 | 0.413 | 0.411 | 0.432 | 0.431 | **0.437** | 0.441 | |
| | | UNI v2 | 0.170 | 0.408 | 0.426 | 0.431 | 0.423 | 0.425 | 0.417 | **0.433** | 0.428 | 0.405 | UNI v1+USD (ER) |
| | | GigaPath | 0.161 | 0.398 | 0.398 | 0.410 | 0.399 | 0.408 | 0.392 | 0.399 | **0.414** | 0.388 | |
| | | ResNet 50 | 0.179 | 0.333 | 0.377 | **0.381** | 0.378 | 0.379 | 0.366 | 0.380 | 0.368 | 0.368 | |
| | Kappa (↑) | UNI v1 | 0.002 | 0.589 | 0.605 | 0.598 | 0.588 | 0.582 | 0.598 | **0.599** | 0.594 | 0.603 | |
| | | UNI v2 | 0.004 | 0.588 | **0.613** | 0.602 | 0.611 | 0.604 | 0.605 | 0.612 | 0.603 | 0.585 | UNI v2+Focal |
| | | GigaPath | 0.000 | 0.568 | 0.574 | **0.590** | 0.576 | 0.582 | 0.564 | 0.570 | 0.577 | 0.567 | |
| | | ResNet 50 | 0.048 | 0.476 | 0.524 | **0.533** | 0.524 | 0.531 | 0.518 | 0.530 | 0.527 | 0.517 | |
| | ECE (↓) | UNI v1 | 0.047 | 0.088 | **0.023** | 0.150 | 0.059 | 0.045 | 0.027 | 0.103 | 0.043 | 0.084 | |
| | | UNI v2 | 0.033 | 0.101 | **0.032** | 0.128 | 0.067 | 0.045 | 0.041 | 0.077 | 0.034 | 0.051 | ResNet 50+LS |
| | | GigaPath | **0.016** | 0.086 | 0.027 | 0.150 | 0.064 | 0.054 | 0.043 | 0.074 | 0.031 | 0.061 | |
| | | ResNet 50 | **0.012** | 0.099 | 0.050 | 0.096 | 0.082 | 0.018 | 0.056 | 0.057 | 0.022 | 0.022 | |

Table 13: Benchmarking average scores under the full metric list for the classification task.

| Datasets | Metrics | Base | Methods | | | | | | | | | | Best Method |
|---|---|---|---|---|---|---|---|---|---|---|---|---|---|
| | | | LS | $L_1$ | Focal | Poly | ER | CE | WER | WPoly | USD (ER) | USD (Poly) | |
| TCGA | ACC | UNI v1 | 0.072 | 0.028 | 0.022 | 0.009 | **0.000** | 0.022 | **0.000** | 0.015 | **0.000** | 0.015 | UNI v2+USD (ER) |
| | | UNI v2 | 0.230 | 0.022 | 0.017 | 0.014 | 0.022 | 0.015 | **0.000** | 0.007 | **0.000** | 0.007 | |
| | | GigaPath | 0.162 | 0.151 | 0.076 | 0.067 | 0.180 | 0.069 | 0.134 | **0.024** | 0.134 | **0.024** | |
| | | ResNet 50 | 0.249 | 0.054 | 0.032 | 0.043 | 0.051 | 0.032 | 0.043 | **0.022** | 0.035 | 0.046 | |
| | AUROC | UNI v1 | 0.149 | 0.010 | 0.010 | 0.009 | 0.008 | 0.012 | **0.005** | 0.009 | **0.005** | 0.009 | UNI v2+USD (ER) |
| | | UNI v2 | 0.264 | **0.001** | **0.000** | **0.000** | 0.005 | **0.000** | **0.000** | **0.000** | **0.000** | **0.000** | |
| | | GigaPath | 0.457 | 0.218 | 0.071 | 0.057 | 0.085 | **0.029** | 0.047 | 0.044 | 0.047 | 0.044 | |
| | | ResNet 50 | 0.320 | 0.061 | 0.024 | 0.019 | 0.022 | 0.016 | 0.014 | **0.012** | 0.027 | 0.032 | |
| | Bacc | UNI v1 | 0.072 | 0.028 | 0.022 | 0.009 | **0.000** | 0.022 | **0.000** | 0.015 | **0.000** | 0.015 | UNI v2+USD (ER) |
| | | UNI v2 | 0.230 | 0.022 | 0.017 | 0.014 | 0.022 | 0.015 | **0.000** | 0.007 | **0.000** | 0.007 | |
| | | GigaPath | 0.162 | 0.151 | 0.076 | 0.067 | 0.180 | 0.069 | 0.134 | **0.024** | 0.134 | **0.024** | |
| | | ResNet 50 | 0.249 | 0.054 | 0.032 | 0.043 | 0.051 | 0.032 | 0.043 | **0.022** | 0.035 | 0.046 | |
| | Kappa | UNI v1 | 0.144 | 0.056 | 0.043 | 0.018 | **0.000** | 0.043 | **0.000** | 0.030 | **0.000** | 0.030 | UNI v2+USD (ER) |
| | | UNI v2 | 0.460 | 0.045 | 0.033 | 0.028 | 0.043 | 0.030 | **0.000** | 0.015 | **0.000** | 0.015 | |
| | | GigaPath | 0.324 | 0.302 | 0.152 | 0.135 | 0.361 | 0.138 | 0.267 | **0.047** | 0.267 | **0.047** | |
| | | ResNet 50 | 0.499 | 0.108 | 0.064 | 0.086 | 0.101 | 0.065 | 0.087 | **0.045** | 0.069 | 0.092 | |
| | wF1 | UNI v1 | 0.074 | 0.028 | 0.022 | 0.009 | **0.000** | 0.022 | **0.000** | 0.015 | **0.000** | 0.015 | UNI v2+USD (ER) |
| | | UNI v2 | 0.250 | 0.022 | 0.017 | 0.014 | 0.022 | 0.015 | **0.000** | 0.008 | **0.000** | 0.008 | |
| | | GigaPath | 0.175 | 0.154 | 0.128 | 0.080 | 0.244 | 0.086 | 0.169 | **0.027** | 0.169 | **0.027** | |
| | | ResNet 50 | 0.272 | 0.101 | 0.034 | 0.045 | 0.090 | 0.038 | 0.067 | **0.023** | 0.047 | 0.048 | |
| | ECE | UNI v1 | 0.057 | 0.016 | 0.027 | 0.013 | 0.012 | 0.024 | 0.024 | **0.011** | 0.024 | **0.011** | UNI v2+Wpoly |
| | | UNI v2 | 0.103 | 0.024 | 0.011 | 0.011 | 0.029 | 0.017 | 0.050 | **0.010** | 0.050 | **0.010** | |
| | | GigaPath | 0.071 | 0.109 | 0.089 | 0.040 | 0.098 | 0.072 | 0.066 | **0.014** | 0.066 | **0.014** | |
| | | ResNet 50 | 0.085 | 0.034 | 0.023 | 0.035 | 0.064 | 0.026 | **0.019** | 0.029 | 0.029 | 0.040 | |
| CAMELYON16 | ACC | UNI v1 | 0.087 | 0.046 | 0.040 | 0.035 | 0.019 | 0.019 | 0.019 | 0.012 | 0.034 | **0.022** | UNI v1+USD (Poly) |
| | | UNI v2 | 0.066 | 0.044 | **0.081** | 0.063 | 0.042 | 0.071 | 0.069 | 0.092 | 0.061 | 0.054 | |
| | | GigaPath | 0.095 | 0.045 | 0.026 | 0.007 | 0.026 | **0.035** | 0.043 | 0.013 | 0.079 | 0.066 | |
| | | ResNet 50 | 0.039 | 0.034 | 0.037 | 0.049 | 0.056 | 0.056 | 0.000 | 0.049 | **0.056** | 0.064 | |
| | AUROC | UNI v1 | 0.112 | 0.026 | 0.047 | 0.019 | 0.025 | 0.026 | 0.013 | 0.024 | 0.026 | **0.020** | ResNet 50+Focal |
| | | UNI v2 | 0.100 | 0.042 | 0.091 | 0.058 | 0.026 | 0.040 | 0.057 | 0.051 | **0.048** | 0.039 | |
| | | GigaPath | 0.097 | **0.076** | 0.063 | 0.035 | 0.096 | 0.068 | 0.062 | 0.052 | 0.089 | 0.055 | |
| | | ResNet 50 | 0.089 | 0.008 | **0.002** | 0.009 | 0.011 | 0.009 | 0.072 | 0.015 | 0.003 | 0.021 | |
| | Bacc | UNI v1 | 0.088 | 0.042 | 0.036 | 0.033 | 0.016 | 0.017 | 0.020 | **0.015** | 0.037 | 0.019 | ResNet 50+ER |
| | | UNI v2 | 0.072 | **0.037** | 0.071 | 0.056 | **0.037** | 0.063 | 0.061 | 0.081 | 0.052 | 0.049 | |
| | | GigaPath | 0.091 | 0.044 | 0.022 | 0.006 | 0.022 | 0.032 | 0.031 | **0.012** | 0.070 | 0.058 | |
| | | ResNet 50 | 0.043 | 0.030 | 0.032 | 0.045 | 0.049 | 0.050 | 0.000 | 0.044 | 0.051 | 0.058 | |
| | Kappa | UNI v1 | 0.174 | 0.083 | 0.072 | 0.065 | 0.034 | 0.035 | 0.038 | **0.027** | 0.069 | 0.040 | ResNet 50+ER |
| | | UNI v2 | 0.141 | 0.071 | 0.137 | 0.105 | **0.070** | 0.119 | 0.115 | 0.154 | 0.099 | 0.092 | |
| | | GigaPath | 0.179 | 0.081 | 0.041 | **0.011** | 0.040 | 0.059 | 0.060 | 0.022 | 0.133 | 0.110 | |
| | | ResNet 50 | 0.043 | 0.032 | 0.062 | 0.084 | 0.093 | 0.050 | 0.000 | 0.044 | 0.051 | 0.058 | |
| | wF1 | UNI v1 | 0.087 | 0.058 | 0.049 | 0.041 | 0.024 | 0.023 | 0.020 | **0.011** | 0.032 | 0.027 | ResNet 50+ER |
| | | UNI v2 | **0.067** | 0.082 | 0.144 | 0.121 | 0.082 | 0.129 | 0.131 | 0.171 | 0.122 | 0.095 | |
| | | GigaPath | 0.149 | 0.092 | 0.067 | **0.014** | 0.092 | 0.071 | 0.111 | 0.028 | 0.143 | 0.126 | |
| | | ResNet 50 | 0.036 | 0.066 | 0.072 | 0.095 | 0.113 | 0.102 | **0.000** | 0.085 | 0.100 | 0.116 | |
| | ECE | UNI v1 | 0.049 | 0.011 | 0.026 | 0.033 | **0.006** | 0.018 | 0.021 | 0.010 | 0.017 | 0.019 | UNI v1+ER |
| | | UNI v2 | 0.061 | **0.022** | 0.038 | 0.066 | 0.036 | 0.054 | 0.047 | 0.062 | 0.043 | **0.022** | |
| | | GigaPath | 0.055 | 0.046 | 0.043 | 0.051 | **0.039** | 0.083 | 0.080 | **0.039** | 0.043 | 0.060 | |
| | | ResNet 50 | 0.026 | 0.037 | **0.009** | 0.044 | 0.059 | 0.017 | 0.025 | 0.014 | 0.047 | 0.046 | |
| PANDA | ACC | UNI v1 | 0.050 | 0.016 | 0.015 | 0.016 | 0.013 | 0.008 | 0.011 | 0.012 | **0.014** | 0.009 | UNI v1+USD (ER) |
| | | UNI v2 | 0.041 | 0.011 | 0.007 | **0.005** | 0.007 | 0.004 | 0.009 | 0.015 | 0.010 | 0.009 | |
| | | GigaPath | 0.014 | 0.009 | 0.012 | 0.013 | 0.017 | 0.003 | 0.009 | 0.006 | **0.006** | 0.007 | |
| | | ResNet 50 | 0.013 | 0.009 | 0.005 | 0.007 | 0.009 | **0.009** | 0.005 | 0.005 | 0.012 | 0.012 | |
| | wF1 | UNI v1 | 0.059 | 0.024 | 0.019 | 0.026 | 0.021 | 0.014 | 0.019 | 0.019 | **0.014** | 0.010 | UNI v1+USD (ER) |
| | | UNI v2 | 0.047 | 0.023 | 0.017 | **0.014** | 0.015 | 0.011 | 0.016 | 0.021 | 0.015 | 0.016 | |
| | | GigaPath | 0.010 | 0.016 | 0.015 | 0.020 | 0.025 | 0.010 | 0.013 | 0.006 | **0.010** | 0.006 | |
| | | ResNet 50 | 0.018 | 0.018 | **0.007** | 0.011 | 0.012 | **0.008** | 0.011 | 0.011 | 0.021 | 0.026 | |
| | Bacc | UNI v1 | 0.012 | 0.021 | 0.013 | 0.023 | 0.016 | 0.013 | 0.017 | 0.017 | 0.016 | **0.008** | GigaPath+Wpoly |
| | | UNI v2 | **0.006** | 0.015 | 0.008 | 0.014 | 0.011 | 0.010 | 0.013 | 0.012 | 0.015 | 0.009 | |
| | | GigaPath | 0.013 | 0.013 | 0.017 | 0.017 | 0.022 | **0.005** | 0.007 | **0.005** | 0.006 | 0.011 | |
| | | ResNet 50 | 0.011 | 0.011 | **0.005** | **0.005** | 0.008 | 0.008 | 0.010 | 0.007 | 0.014 | 0.021 | |
| | Kappa | UNI v1 | 0.027 | 0.022 | 0.011 | 0.021 | 0.013 | 0.021 | 0.013 | 0.018 | 0.013 | **0.005** | UNI v1+USD (Poly) |
| | | UNI v2 | 0.025 | 0.018 | 0.011 | **0.007** | 0.010 | 0.012 | 0.013 | 0.010 | 0.017 | 0.008 | |
| | | GigaPath | 0.045 | 0.013 | 0.019 | 0.019 | 0.021 | 0.011 | 0.012 | **0.007** | 0.010 | 0.011 | |
| | | ResNet 50 | 0.031 | 0.029 | 0.017 | **0.009** | 0.012 | 0.014 | 0.020 | 0.013 | 0.013 | 0.034 | |
| | ECE | UNI v1 | 0.016 | 0.015 | **0.005** | 0.034 | 0.009 | 0.014 | 0.011 | 0.013 | 0.019 | 0.008 | ResNet 50+USD (ER) |
| | | UNI v2 | 0.017 | 0.014 | 0.014 | 0.030 | 0.011 | 0.009 | 0.017 | 0.037 | 0.010 | **0.007** | |
| | | GigaPath | 0.012 | 0.017 | **0.008** | 0.018 | 0.012 | 0.020 | 0.011 | 0.034 | 0.013 | 0.019 | |
| | | ResNet 50 | 0.013 | 0.011 | 0.005 | 0.011 | 0.012 | **0.004** | 0.015 | 0.009 | **0.004** | 0.009 | |

Table 14: Benchmarking standard deviation under the full metric list for the classification task.

| Metrics | Methods | HEST-1K | | | | | | STImage-1K4M | |
| --- | --- | --- | --- | --- | --- | --- | --- | --- | --- |
| | | IDC | READ | PRAD | LYMPH_IDC | COAD | CCRCC | Brain | Skin |
| SPCC | MSE | 0.009 | 0.007 | **0.000** | 0.074 | 0.007 | 0.004 | 0.001 | 0.010 |
| | Huber | 0.006 | 0.010 | 0.003 | 0.008 | **0.001** | 0.002 | **0.001** | 0.015 |
| | PCCMSE | 0.005 | 0.006 | 0.002 | 0.008 | **0.001** | 0.002 | **0.001** | 0.021 |
| | USD (er) | 0.005 | 0.005 | 0.003 | 0.036 | **0.002** | 0.006 | **0.002** | 0.007 |
| GPCC | MSE | 0.008 | 0.003 | **0.002** | 0.138 | 0.005 | 0.005 | 0.003 | 0.027 |
| | Huber | 0.004 | 0.004 | 0.003 | 0.003 | **0.002** | 0.008 | 0.004 | 0.066 |
| | PCCMSE | **0.002** | 0.003 | 0.005 | **0.002** | 0.002 | 0.003 | 0.005 | 0.008 |
| | USD (er) | 0.003 | 0.004 | 0.008 | 0.026 | **0.002** | 0.008 | **0.002** | 0.026 |
| MSE | MSE | 0.073 | 0.004 | **0.001** | 0.035 | 0.013 | **0.001** | 0.001 | 0.011 |
| | Huber | 0.065 | 0.002 | 0.002 | 0.005 | 0.009 | **0.001** | 0.001 | 0.035 |
| | PCCMSE | 0.025 | 0.003 | 0.002 | 0.007 | 0.002 | **0.000** | 0.002 | 0.016 |
| | USD (er) | 0.069 | 0.018 | 0.004 | 0.022 | 0.006 | 0.007 | **0.003** | 0.020 |

Table 15: Benchmarking standard deviation for the full metric list based on the regression task.

KMeans clustering after processing the data with Harmony, and the image features are extracted with UNI v2. Table 22 shows that selecting the best k based on tuning ASW score achieves the highest SPCC score in over 75% datasets from both the HEST and STImage1k4M databases, and its GPCC and MSE are also in the top2 list for most of the datasets. Moreover, using the best k can obviously reduce the randomness and improve training robustness evaluated with all three metrics, especially in the IDC and LYMPH_IDC datasets, since the results based on k=7 and 11 for IDC and k=5 and 7 for Brain show high variance in the evaluation with MSE or SPCC across five random seeds. Therefore, tuning the cluster number k with ASW score is an effective approach to select the size used for model training, supported by its superiority in average performance and robustness.

### 8.13 COMPARISONS BETWEEN USD (ER) AND USD (POLY) FOR THE GENE EXPRESSION PREDICTION.

The results for comparing two modes of USD are shown in Figure 8. According to this figure, these two modes do not show obvious differences across all selected metrics.

### 8.14 COMPARISONS BETWEEN DIFFERENT BASE MODELS FOR GENE EXPRESSION PREDICTION

According to Tables 23 and 24, UNI v2-based combination always outperforms other combinations evaluated by GPCC, and it also has low variance. Therefore, UNI v2 is selected as the base model for evaluating the performances of gene expression prediction based on different training strategies.

### 8.15 BROADER IMPACT AND LIMITATIONS

One possible limitation of USD could be the task-specific requirements of pathology foundation models, as the sample difficulty is affected by the source representations, and thus different foundation models might lead to differences in estimating sample difficulty. One potential solution is to define a metric to select models before estimating sample difficulty. The other limitation could be training efficiency for large-scale datasets, which could potentially be addressed by using advanced GPU cores.

| Datasets | Metrics | Base | Methods | |
| | | | LR | USD (ER) |
|---|---|---|---|---|
| TCGA | ACC | UNI v1 | 0.733 | 0.933 |
| | | UNI v2 | 0.917 | 1.000 |
| | | GigaPath | 0.617 | 0.677 |
| | | ResNet 50 | 0.500 | 0.627 |
| | AUROC | UNI v1 | 0.994 | 0.994 |
| | | UNI v2 | 1.000 | 1.000 |
| | | GigaPath | 0.919 | 0.898 |
| | | ResNet 50 | 0.023 | 0.677 |
| | Bacc | UNI v1 | 0.733 | 0.933 |
| | | UNI v2 | 0.917 | 1.000 |
| | | GigaPath | 0.617 | 0.677 |
| | | ResNet 50 | 0.500 | 0.627 |
| | Kappa | UNI v1 | 0.467 | 0.867 |
| | | UNI v2 | 0.833 | 1.000 |
| | | GigaPath | 0.233 | 0.353 |
| | | ResNet 50 | 0.000 | 0.253 |
| | wF1 | UNI v1 | 0.713 | 0.933 |
| | | UNI v2 | 0.916 | 1.000 |
| | | GigaPath | 0.551 | 0.627 |
| | | ResNet 50 | 0.333 | 0.613 |
| | ECE | UNI v1 | 0.329 | 0.065 |
| | | UNI v2 | 0.329 | 0.151 |
| | | GigaPath | 0.236 | 0.208 |
| | | ResNet 50 | 0.263 | 0.184 |
| CAMELYON16 | ACC | UNI v1 | 0.618 | 0.741 |
| | | UNI v2 | 0.618 | 0.562 |
| | | GigaPath | 0.647 | 0.491 |
| | | ResNet 50 | 0.500 | 0.585 |
| | AUROC | UNI v1 | 0.756 | 0.812 |
| | | UNI v2 | 0.751 | 0.753 |
| | | GigaPath | 0.716 | 0.643 |
| | | ResNet 50 | 0.672 | 0.730 |
| | Bacc | UNI v1 | 0.620 | 0.757 |
| | | UNI v2 | 0.638 | 0.592 |
| | | GigaPath | 0.655 | 0.531 |
| | | ResNet 50 | 0.541 | 0.617 |
| | Kappa | UNI v1 | 0.237 | 0.496 |
| | | UNI v2 | 0.264 | 0.173 |
| | | GigaPath | 0.303 | 0.060 |
| | | ResNet 50 | 0.074 | 0.219 |
| | wF1 | UNI v1 | 0.618 | 0.750 |
| | | UNI v2 | 0.599 | 0.498 |
| | | GigaPath | 0.646 | 0.350 |
| | | ResNet 50 | 0.376 | 0.524 |
| | ECE | UNI v1 | 0.251 | 0.107 |
| | | UNI v2 | 0.220 | 0.159 |
| | | GigaPath | 0.227 | 0.199 |
| | | ResNet 50 | 0.181 | 0.137 |
| PANDA | ACC | UNI v1 | 0.458 | 0.495 |
| | | UNI v2 | 0.448 | 0.488 |
| | | GigaPath | 0.449 | 0.473 |
| | | ResNet 50 | 0.378 | 0.430 |
| | wF1 | UNI v1 | 0.448 | 0.479 |
| | | UNI v2 | 0.438 | 0.488 |
| | | GigaPath | 0.439 | 0.455 |
| | | ResNet 50 | 0.307 | 0.397 |
| | Bacc | UNI v1 | 0.408 | 0.437 |
| | | UNI v2 | 0.398 | 0.428 |
| | | GigaPath | 0.405 | 0.414 |
| | | ResNet 50 | 0.280 | 0.368 |
| | Kappa | UNI v1 | 0.569 | 0.594 |
| | | UNI v2 | 0.551 | 0.603 |
| | | GigaPath | 0.556 | 0.577 |
| | | ResNet 50 | 0.286 | 0.527 |
| | ECE | UNI v1 | 0.175 | 0.043 |
| | | UNI v2 | 0.206 | 0.034 |
| | | GigaPath | 0.142 | 0.031 |
| | | ResNet 50 | 0.045 | 0.022 |

Table 16: Evaluation results between LR and USD (ER).

| Metircs | Methods | Datasets and Statistics | | | | | | | | |
|---------|---------|------|------|------|-----------|------|-------|-------|-------|---------|
| | | IDC | READ | PRAD | LYMPH_IDC | COAD | CCRCC | Brain | Skin | Average |
| SPCC | DeepPT | 0.579 | 0.314 | 0.688 | 0.084 | 0.611 | 0.379 | 0.603 | 0.389 | 0.456 |
| | USD (ER) | **0.589** | **0.381** | **0.689** | **0.129** | **0.622** | **0.386** | **0.618** | **0.409** | **0.478** |
| GPCC | DeepPT | 0.386 | 0.186 | 0.110 | 0.223 | 0.563 | 0.263 | 0.110 | 0.189 | 0.254 |
| | USD (ER) | **0.400** | **0.283** | **0.138** | **0.236** | **0.565** | **0.273** | **0.154** | **0.265** | **0.289** |
| MSE | DeepPT | 2.947 | 0.277 | 0.296 | 0.868 | 0.986 | **0.491** | 0.282 | 1.668 | 0.977 |
| | USD (ER) | **2.754** | **0.269** | **0.294** | **0.857** | **0.957** | 0.492 | **0.279** | **1.481** | **0.923** |

Table 17: Comparing average scores between DeepPT and USD for the gene expression prediction task.

| Metircs | Methods | Datasets and Statistics | | | | | | | | |
|---------|---------|------|------|------|-----------|------|-------|-------|-------|---------|
| | | IDC | READ | PRAD | LYMPH_IDC | COAD | CCRCC | Brain | Skin | Average |
| SPCC | DeepPT | **0.004** | **0.003** | **0.001** | **0.025** | 0.003 | **0.005** | 0.003 | 0.014 | **0.007** |
| | USD (ER) | 0.005 | 0.005 | 0.003 | 0.036 | **0.002** | 0.006 | **0.002** | **0.007** | 0.008 |
| GPCC | DeepPT | 0.004 | **0.003** | **0.006** | **0.026** | 0.004 | 0.013 | 0.013 | **0.015** | 0.011 |
| | USD (ER) | **0.003** | 0.004 | 0.008 | **0.026** | **0.002** | **0.008** | **0.002** | 0.026 | **0.010** |
| MSE | DeepPT | **0.027** | **0.002** | **0.001** | **0.015** | 0.009 | **0.001** | **0.001** | **0.016** | **0.009** |
| | USD (ER) | 0.069 | 0.018 | 0.004 | 0.022 | **0.006** | 0.007 | 0.003 | 0.020 | 0.019 |

Table 18: Comparing standard deviation between DeepPT and USD for the gene expression prediction task.

| Metrics | FPT | TPH |
|---------|-----|-----|
| Acc | 0.487 (0.014) | 0.912 (0.020) |
| AUROC | 0.600 (0.236) | 0.984 (0.010) |
| Bacc | 0.487 (0.014) | 0.900 (0.038) |
| Kappa | -0.028 (0.028) | 0.827 (0.043) |
| wF1 | 0.327 (0.006) | 0.913 (0.022) |

Table 19: Performances of two modes for training the prediction head. The format of value in the table is: average (standard deviation).

| Datasets | Metrics | Methods | | | |
|----------|---------|---------------|---------------|---------------|---------------|
| | | MP | RD | ABMIL | Default |
| CAMELYON16 | ACC | 0.524 (0.035) | 0.562 (0.048) | 0.509 (0.017) | 0.756 (0.022) |
| | AUROC | 0.643 (0.060) | 0.641 (0.048) | 0.538 (0.018) | 0.834 (0.020 |
| | Bacc | 0.542 (0.026) | 0.580 (0.055) | 0.071 (0.033) | 0.771 (0.019) |
| | Kappa | 0.082 (0.052) | 0.152 (0.103) | 0.448 (0.037) | 0.525 (0.040) |
| | wF1 | 0.496 (0.060) | 0.537 (0.052) | 0.618 (0.059) | 0.750 (0.027) |
| | ECE | 0.212 (0.033) | 0.188 (0.046) | 0.267 (0.039) | 0.107 (0.019) |

Table 20: Performances of four different strategies for training the prediction head. The format of value in the table is: average (standard deviation).

| Datasets | Metrics | Methods | |
|---|---|---|---|
| | | $\mathcal{MRMD}$ (base) | $\mathcal{MRMD}$ (class-removal) |
| CAMELYON16 | ACC | 0.756 (0.022) | 0.750 (0.021) |
| | AUROC | 0.834 (0.020 | 0.831 (0.009) |
| | Bacc | 0.771 (0.019) | 0.767 (0.018) |
| | Kappa | 0.525 (0.040) | 0.514 (0.038) |
| | wF1 | 0.750 (0.027) | 0.743 (0.025) |
| | ECE | 0.107 (0.019) | 0.109 (0.154) |

Table 21: Performances of two different strategies for computing sample difficulty. The format of value in the table is: average (standard deviation).

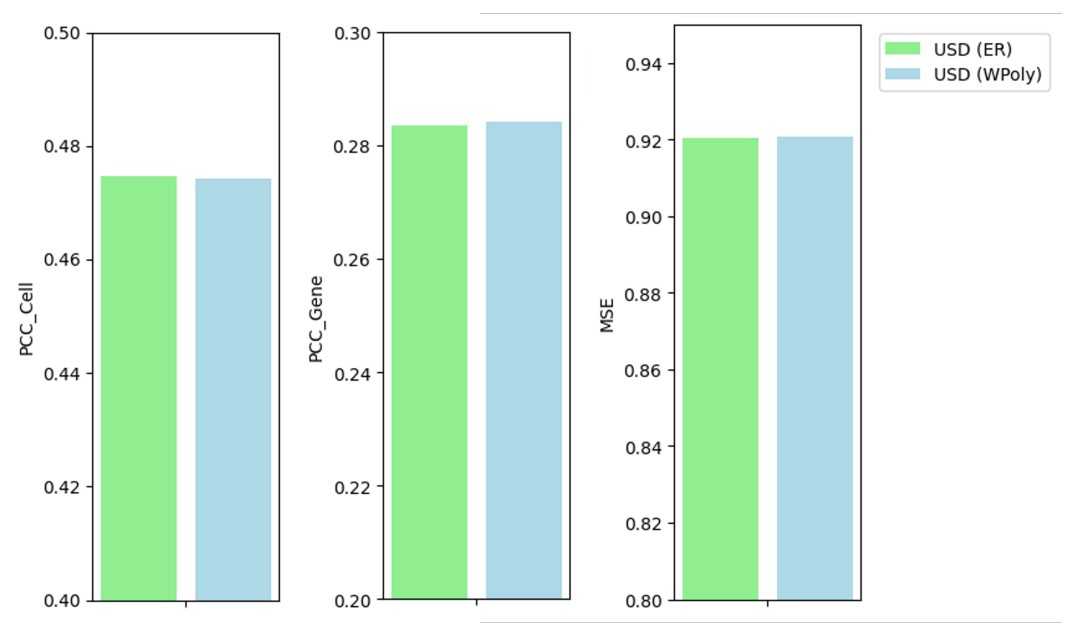

Figure 8: Benchmarking scores averaged by all tested datasets between USD (ER) and USD (Poly).

| Dataset (best k) | Metric | k=3 | k=5 | k=7 | k=9 | k=11 |
|---|---|---|---|---|---|---|
| IDC (best k=3) | SPCC | 0.591 (0.004) | 0.590 (0.003) | 0.591 (0.007) | 0.591 (0.008) | 0.589 (0.008) |
| | GPCC | 0.400 (0.003) | 0.401 (0.004) | 0.396 (0.004) | 0.403 (0.002) | 0.398 (0.008) |
| | MSE | 2.69 (0.005) | 2.70 (0.083) | 2.80 (0.173) | 2.75 (0.067) | 2.76 (0.150) |
| READ (best k=3) | SPCC | 0.381 (0.005) | 0.383 (0.005) | 0.379 (0.006) | 0.379 (0.004) | 0.383 (0.010) |
| | GPCC | 0.283 (0.004) | 0.285 (0.001) | 0.282 (0.002) | 0.286 (0.005) | 0.285 (0.004) |
| | MSE | 0.269 (0.018) | 0.270 (0.012) | 0.265 (0.009) | 0.267 (0.011) | 0.270 (0.004) |
| PRAD (best k=3) | SPCC | 0.690 (0.003) | 0.687 (0.003) | 0.686 (0.002) | 0.686 (0.004) | 0.689 (0.003) |
| | GPCC | 0.138 (0.008) | 0.134 (0.006) | 0.132 (0.004) | 0.136 (0.009) | 0.138 (0.003) |
| | MSE | 0.294 (0.004) | 0.293 (0.004) | 0.295 (0.002) | 0.294 (0.003) | 0.291 (0.002) |
| LYMPH_IDC (k=7) | SPCC | 0.119 (0.051) | 0.120 (0.049) | 0.129 (0.036) | 0.136 (0.046) | 0.101 (0.044) |
| | GPCC | 0.241 (0.020) | 0.239 (0.018) | 0.236 (0.026) | 0.250 (0.006) | 0.205 (0.052) |
| | MSE | 0.864 (0.025) | 0.862 (0.019) | 0.857 (0.022) | 0.872 (0.020) | 0.852 (0.023) |
| COAD (best k=3) | SPCC | 0.622 (0.002) | 0.625 (0.002) | 0.625 (0.003) | 0.624 (0.005) | 0.623 (0.004) |
| | GPCC | 0.565 (0.002) | 0.567 (0.003) | 0.568 (0.002) | 0.567 (0.002) | 0.565 (0.004) |
| | MSE | 0.957 (0.006) | 0.958 (0.003) | 0.959 (0.005) | 0.953 (0.009) | 0.959 (0.010) |
| CCRC (best k=3) | SPCC | 0.386 (0.006) | 0.383 (0.003) | 0.383 (0.007) | 0.386 (0.011) | 0.382 (0.006) |
| | GPCC | 0.273 (0.008) | 0.274 (0.005) | 0.272 (0.007) | 0.273 (0.008) | 0.270 (0.007) |
| | MSE | 0.492 (0.007) | 0.492 (0.004) | 0.494 (0.001) | 0.493 (0.005) | 0.495 (0.004) |
| Brain (best k=3) | SPCC | 0.618 (0.002) | 0.613 (0.008) | 0.610 (0.008) | 0.610 (0.007) | 0.612 (0.007) |
| | GPCC | 0.154 (0.002) | 0.157 (0.007) | 0.155 (0.008) | 0.157 (0.010) | 0.161 (0.010) |
| | MSE | 0.279 (0.004) | 0.279 (0.004) | 0.280 (0.005) | 0.281 (0.005) | 0.276 (0.003) |
| Skin (best k=3) | SPCC | 0.409 (0.007) | 0.395 (0.021) | 0.390 (0.023) | 0.396 (0.013) | 0.409 (0.009) |
| | GPCC | 0.265 (0.020) | 0.255 (0.019) | 0.262 (0.019) | 0.255 (0.020) | 0.258 (0.025) |
| | MSE | 1.481 (0.020) | 1.580 (0.014) | 1.572 (0.013) | 1.589 (0.025) | 1.581 (0.010) |

Table 22: Effect of cluster number $k$ with format score (standard deviation) for the regression task.

| Metrics | Methods | Base Models | | | |
|---|---|---|---|---|---|
| | | UNI v1 | UNI v2 | GigaPath | ResNet 50 |
| SPCC | MSE Loss | 0.572 | 0.581 | **0.588** | 0.550 |
| | Huber Loss | 0.562 | **0.589** | 0.585 | 0.527 |
| | PCCMSE Loss | 0.575 | 0.588 | **0.599** | 0.554 |
| GPCC | MSE Loss | 0.348 | **0.389** | 0.330 | 0.267 |
| | Huber Loss | 0.334 | **0.390** | 0.321 | 0.220 |
| | PCCMSE Loss | 0.355 | **0.400** | 0.351 | 0.281 |
| MSE | MSE Loss | 3.424 | 2.825 | 2.945 | **2.598** |
| | Huber Loss | 3.512 | **2.812** | 2.951 | 2.919 |
| | PCCMSE Loss | 3.414 | 2.748 | 2.855 | **2.561** |

Table 23: Benchmarking average scores for the full metric list based on different base models for the regression task.

| Metrics | Methods | Base Models | | | |
|---|---|---|---|---|---|
| | | UNI v1 | UNI v2 | GigaPath | ResNet 50 |
| SPCC | MSE Loss | 0.013 | 0.009 | **0.004** | 0.006 |
| | Huber Loss | 0.010 | **0.006** | **0.006** | 0.007 |
| | PCCMSE Loss | 0.006 | 0.005 | 0.005 | **0.002** |
| GPCC | MSE Loss | **0.002** | 0.008 | 0.017 | 0.019 |
| | Huber Loss | 0.008 | **0.004** | 0.006 | 0.016 |
| | PCCMSE Loss | 0.003 | **0.002** | 0.007 | 0.003 |
| MSE | MSE Loss | **0.034** | 0.073 | 0.059 | 0.043 |
| | Huber Loss | **0.026** | 0.065 | 0.075 | 0.079 |
| | PCCMSE Loss | 0.030 | **0.025** | 0.032 | 0.031 |

Table 24: Benchmarking standard deviation for the full metric list based on different base models for the regression task.

