# OpenReview forum: "Learning Universal Sample Difficulty with Pathology Foundation Models in Histopathology Image Analysis"
_ICLR.cc/2026/Conference — ICLR 2026 Conference Withdrawn Submission_

### Official Review · Reviewer_guJe · 2025-10-21

**Soundness:** 3
**Presentation:** 3
**Contribution:** 2
**Rating:** 4
**Confidence:** 4

**Summary:**

This paper proposes a framework to estimate sample difficulty from PFM embeddings using a modified relative Mahalanobis distance (MRMD). Samples consistently misclassified by LR are assigned maximal difficulty. Difficulty weights regularize training (either entropy-regularized CE or Poly) for classification. The proposed method improves over baselines across 3 classification datasets.

**Strengths:**

1.	Clear high-level idea: Estimating difficulty in representation space and using it to weight training is intuitive; extending to regression via discretization is practical.
2.	Breadth of evaluation: Multiple datasets (including large histopathology datasets and 8 spatial transcriptomics cohorts), several PFMs (UNI v1/v2, GigaPath, plus ResNet50), and a reasonably comprehensive set of baselines and ablations (choice of K, batch correction, loss components, training head vs full finetune).

**Weaknesses:**

1.	MRMD definition/sign: The paper defines MD with a negative quadratic form and then $RMD = MD_k − MD_b$, which is counter to the conventional “distance ≥ 0” notion. It’s not fully clear whether larger MRMD corresponds to harder or easier samples before normalization.
2.	Gaussian modeling: Fitting full covariances $Σ$ and $Σ_b$ in high-dimensional PFM embedding spaces can be ill-conditioned given typical sample sizes. No discussion of regularization/shrinkage or dimensionality reduction prior to MD computation is provided. This may be crucial for robustness and reproducibility.
3.	Regression formulation and OE term: In Eq. (9), the second term collapses to zero because each feature is its own center, leaving only pairwise penalties between “centers.” This deviates from OE formulations that trade off tightness and diversity. More justification is needed.
4.	Stroger difficulty estimation baselines: Compare against stronger baselines such as kNN density in embedding space, energy-based OOD scores, ODIN/Mahalanobis OOD, deep ensemble/MC-dropout uncertainties, and sample reweighting via OHEM or margin-based hardness.

**Questions:**

See Weaknesses.

---

### Official Review · Reviewer_unxz · 2025-10-30

**Soundness:** 2
**Presentation:** 1
**Contribution:** 2
**Rating:** 2
**Confidence:** 5

**Summary:**

This paper investigates sample difficulty in histology image analysis, which is estimated by prior-informed relative Mahalanobis distance (RMD), for both classification and regression tasks. For those difficult samples, a higher weight is assigned to regularize the training.

**Strengths:**

1. It is new to explore sample difficulty in histology image analysis.
2. Both classification and regression are considered.

**Weaknesses:**

1. The elaboration in this paper is ambiguous. For example, it is confusing about low-level information. What’s the low-level information? What’s a low-low-calibration level? What’s the relationship between “current models tend to be overconfident” and sample difficulty?
2. Some explanations are inaccurate. TCGA LUSC-LUAD is for cancer subtyping, and CAMELYON16 is for the detection of cancer metastasis instead of the disease state. PANDA is for grading, which is actually supposed to be measured by the kappa coefficient.
3. The novelty is incremental. The authors simply applied RMD [1] in slide-level or patch-level embedding without sufficient motivation.
4. The empirical evidence cannot support that learning based on sample difficulty is necessary. The common pipeline of modeling WSI is multiple instance learning. However, no comparison to these MIL approaches is given.

[1] Learning Sample Difficulty from Pre-trained Models for Reliable Prediction, NeurIPS 2023.

**Questions:**

See weakness.

---

### Official Review · Reviewer_WQsu · 2025-10-31

**Soundness:** 2
**Presentation:** 1
**Contribution:** 2
**Rating:** 2
**Confidence:** 3

**Summary:**

This paper introduces a method to quantify the difficulty of training samples, in order to increase their importance in the training loss, with a specific focus on the digital pathology data domain. The difficulty of a sample is derived from its similarity to other samples from the same label and ones from other labels in embedding space, but also from errors made by the trained model on the given sample. The method is compared against diverse baselines on different slide-level histopathology datasets.

**Strengths:**

- S1:  Estimating sample difficulty is a promising direction to improve performance and calibration of classifiers.
- S2: The method is compared against many other baselines on diverse datasets.

**Weaknesses:**

- W1: [Major] The paper lacks clarity, making it hard to properly understand the contributions. It is thus very challenging to judge its relevance for the community. The introduction does not tell a clear story, and the methods section is hard to follow. This explains why my summary of this paper is rather short. I’m ready to re-consider this point if other reviewers have an opposite impression, but clarity was a major obstacle for me to properly assess the quality of the presented work.
- W2: [Major] It is surprising that different methods achieve 100% accuracy and AUROC on the TCGA dataset (Table 1). Do authors have any reflexion about this?
- W3: [Major] The authors present their method in the context of histopathology image analysis. However, it is not clear to me whether digital pathology is a better fit for this contribution than any other domain. Could authors elaborate on this?
- W4: [Minor] L.200 {x_i} is used to refer to features while it was previously used to refer to the set of images. A feature-specific notations should be introduced here.
- W5: [Minor] Many paper references should be between parentheses (you can use “\citep{}” in latex).

**Questions:**

- Q1: [Related to W1] Can authors explain how they will improve the presentation of their method and overall contributions to make the paper easier to follow?
- Q2: [Related to W2] Do authors have any reflexion about the 100% accuracy of many baselines on the TCGA dataset (Table 1)?
- Q3: [Related to W3] Could authors explain why histopathology is more relevant as the application domain for this method compared with other domains?

---

### Official Review · Reviewer_88i4 · 2025-11-01

**Soundness:** 3
**Presentation:** 2
**Contribution:** 2
**Rating:** 4
**Confidence:** 4

**Summary:**

The paper improves downstream task learning with pathology foundation models by estimating sample difficulty in an unsupervised fashion and then using these difficulty weights to assign larger weights to more difficult samples in the loss. The authors perform experiments with three publicly available foundation models (UNI, UNIv2, Gigapath) and one ResNet-50 ImageNet baseline on five different benchmarks (TCGA subset, CAMELYON16, PANDA, HEST-1k, STImage-1K4M). Their experiments showed that the method can improve the accuracy on morphology based classification tasks  by up to 3.8% and gene-level correlation by up to 62.2%.

**Strengths:**

* Improving the downstream probes for pathology foundation models is an understudied problem. In that regard the paper makes a valuable contribution.

* The evaluation is performed over multiple public datasets and publicly available models. Therefore, the results can be easily reproduced.

* The downstream training is compared to multiple baseline methods (LS, L_1, Focal, Poly, WER, WPoly).

* The method archives significant gains over the standard training regimes.

**Weaknesses:**

* The paper basically just applies the method from [1] to the pathology domain. It is interesting how much impact it can have there but the methodological novelty is limited.

* The paper uses the Malahanobis distance to measure the outlierness of a sample relative to the other samples in the class. Only the Malahanobis distance is considered and not any other outlier/anomaly detection methods. For an applied study, it would be particularly interesting to compare various methods from the literature and see how they are performing. Further, it would be good to put the paper into context of other anomaly detection works on pathology images.

* Stylistic: The citations do not use brackets which make the paper really hard to read in some paragraphs. Additionally, some of the plot axis values (yticks) are really hard to read (e.g. Fig 4).

[1] Peng Cui, Dan Zhang, Zhijie Deng, Yinpeng Dong, and Jun Zhu. Learning sample difficulty from pre-trained models for reliable prediction. Advances in Neural Information Processing Systems, 36:25390–25408, 2023.

**Questions:**

* Why are not all  HEST-1K tasks used? The following are missing: PAAD, SKCM, LUAD. This always looks a little bit suspicious. Also for the other tasks standardized splits from a commonly used evaluation framework for pathology foundation models ( https://github.com/kaiko-ai/eva/tree/main) could be used. This would make the paper much stronger in my view.

* How were the hyperparameters for the proposed and all the baseline methods tuned (How many values were considered and how was the validation split done)?

---

### Note · Authors · 2025-11-12

**Comment:**

We will withdraw this manuscript.

**Withdrawal Confirmation:**

I have read and agree with the venue's withdrawal policy on behalf of myself and my co-authors.